# Nicotinamide mononucleotide adenylyltransferase uses its NAD⁺ substrate-binding site to chaperone phosphorylated Tau

Xiaojuan Ma[1,2†], Yi Zhu[3†], Jinxia Lu[4,5], Jingfei Xie[1,2], Chong Li[3‡], Woo Shik Shin[6], Jiali Qiang[1,2], Jiaqi Liu[7], Shuai Dou[1], Yi Xiao[1], Chuchu Wang[1,2], Chunyu Jia[1,2], Houfang Long[1,2], Juntao Yang[8], Yanshan Fang[1], Lin Jiang[6], Yaoyang Zhang[1], Shengnan Zhang[1], Rong Grace Zhai[3*], Cong Liu[1*], Dan Li[4,5*]

[1]Interdisciplinary Research Center on Biology and Chemistry, Shanghai Institute of Organic Chemistry, Chinese Academy of Sciences, Shanghai, China; [2]University of the Chinese Academy of Sciences, Beijing, China; [3]Department of Molecular and Cellular Pharmacology, University of Miami Miller School of Medicine, Miami, United States; [4]Bio-X-Renji Hospital Research Center, Renji Hospital, School of Medicine, Shanghai Jiao Tong University, Shanghai, China; [5]Bio-X Institutes, Key Laboratory for the Genetics of Developmental and Neuropsychiatric Disorders, Ministry of Education, Shanghai Jiao Tong University, Shanghai, China; [6]Department of Neurology, Molecular Biology Institute, and Brain Research Institute, University of California, Los Angeles, Los Angeles, United States; [7]School of Pharmacy, Key Laboratory of Molecular Pharmacology and Drug Evaluation (Yantai University), Ministry of Education, Collaborative Innovation Center of Advanced Drug Delivery System and Biotech Drugs in Universities of Shandong, Yantai University, Yantai, China; [8]State Key Laboratory of Medical Molecular Biology, Department of Biochemistry and Molecular Biology, Institute of Basic Medical Sciences, Chinese Academy of Medical Sciences & Peking Union Medical College, Beijing, China

*For correspondence:
gzhai@miami.edu (RGZ);
liulab@sioc.ac.cn (CL);
lidan2017@sjtu.edu.cn (DL)

†These authors contributed equally to this work

Present address: ‡Institute of Molecular Biotechnology of the Austrian Academy of Science (IMBA), Vienna, Austria

Competing interests: The authors declare that no competing interests exist.

**Abstract** Tau hyper-phosphorylation and deposition into neurofibrillary tangles have been found in brains of patients with Alzheimer's disease (AD) and other tauopathies. Molecular chaperones are involved in regulating the pathological aggregation of phosphorylated Tau (pTau) and modulating disease progression. Here, we report that nicotinamide mononucleotide adenylyltransferase (NMNAT), a well-known NAD⁺ synthase, serves as a chaperone of pTau to prevent its amyloid aggregation in vitro as well as mitigate its pathology in a fly tauopathy model. By combining NMR spectroscopy, crystallography, single-molecule and computational approaches, we revealed that NMNAT adopts its enzymatic pocket to specifically bind the phosphorylated sites of pTau, which can be competitively disrupted by the enzymatic substrates of NMNAT. Moreover, we found that NMNAT serves as a co-chaperone of Hsp90 for the specific recognition of pTau over Tau. Our work uncovers a dedicated chaperone of pTau and suggests NMNAT as a key node between NAD⁺ metabolism and Tau homeostasis in aging and neurodegeneration.

## Introduction

Phosphorylated Tau (pTau) is the major component of the neurofibrillary tangles that are commonly found in the brains of patients with Alzheimer's disease (AD) and many other tauopathy-related

neurodegenerative diseases (*Avila, 2006*; *Hanger et al., 2009*; *Hanger et al., 2002*; *Hanger et al., 1991*). Tau is an intrinsically disordered protein with a high abundance in neurons (*Hirokawa et al., 1996*; *Konzack et al., 2007*). There are six isoforms of Tau in the human central nervous system due to alternative splicing (*Goedert et al., 1989*). Under the physiological condition, Tau associates with microtubules and modulates the stability of axonal microtubules (*Drechsel et al., 1992*). Whereas, phosphorylation of Tau by protein kinases such as microtubule affinity regulating kinase 2 (MARK2), causes the release of Tau from microtubule binding, which leads to hyper-phosphorylation and amyloid aggregation of Tau (*Ando et al., 2016*; *Biernat et al., 1993*; *Drewes, 2004*; *Drewes et al., 1997*; *Gu et al., 2013*). The amyloid aggregation of Tau is closely associated with the pathogenesis of AD and other tauopathies (*Drewes et al., 1997*). Different proteins including chaperones (Hsp90, Hsc70/Hsp70) (*Dickey et al., 2007a*), proteasome (*Dickey et al., 2007b*) and protein phosphatase 2A (PP2A) (*Gong et al., 1993*) were found to play important roles in maintaining Tau homeostasis including preventing abnormal hyper-phosphorylation and aggregation, and facilitating pTau degradation (*Hanger et al., 2009*).

Nicotinamide mononucleotide adenylyltransferase (NMNAT) was initially identified as an NAD$^+$ synthase that catalyzes the reversible conversion of NMN (nicotinamide mononucleotide) to NAD$^+$ in the final step of both the de novo biosynthesis and salvage pathways in most organisms across all three kingdoms of life (*Magni et al., 1999*). NMNAT is indispensable in maintaining neuronal homeostasis (*Araki, 2004*; *Conforti et al., 2009*). Familial mutations of NMNAT have been found to cause Leber congenital amaurosis 9 (LCA9) (*Chiang et al., 2012*; *Falk et al., 2012*; *Koenekoop et al., 2012*; *Perrault et al., 2012*) and retinal degeneration (*Beirowski et al., 2008*). Moreover, NMNAT is closely related to AD and other tauopathies. The mRNA level of human NMNAT2 (hN2), one of the three isoforms of human NMNATs (*Raffaelli et al., 2002*), decreases in patients of AD (*Ali et al., 2016*). Abundant hN2 proteins were detected in the insoluble brain fraction of AD patients, which also contains pTau and Hsp90 (*Ali et al., 2016*). In addition, NMNAT plays a protective role in different cellular and animal models of AD (*Ali et al., 2013*; *Conforti et al., 2014*; *Fang et al., 2012*; *Ocampo et al., 2013*). Over-expression of different isoforms of NMNAT can significantly reduce the abnormal aggregation (*Zhai et al., 2008*) and cytotoxicity of pTau and relieve pTau burden in different models of AD (*Rossi et al., 2018*) and frontotemporal dementia with parkinsonism linked to chromosome 17 (FTDP-17) (*Ali et al., 2012*; *Ljungberg et al., 2012*).

Intriguingly, in addition to the well-studied NAD$^+$ synthase activity, NMNAT has been found to be able to retrieve the activity of luciferase from heat-denatured amorphous aggregation suggesting a chaperone-like activity of NMNAT (*Ali et al., 2016*; *Zhai et al., 2008*). However, it remains puzzling how a single domain enzyme, that has no similarity to any known chaperones, fulfills a chaperone-like activity. Moreover, it is confusing whether the protective role of NMNAT in the AD animal models comes from its chaperone-like activity or enzymatic activity, given that NAD$^+$ is an essential cofactor in cellular processes such as transcriptional regulation (*D'Amours et al., 1998*; *Shogren-Knaak et al., 2006*; *Zhang et al., 2009*) and oxidative reactions (*Lewis et al., 2014*).

In this work, we demonstrate the chaperone-like activity of NMNAT against the amyloid aggregation of Tau in vitro and in the fly model. By combining multiple biophysical and computational approaches, we reveal the molecular mechanism of NMNAT as a specific chaperone of pTau. Our work provides the structural basis for how NMNAT manages its dual functions as both an enzyme and a chaperone, as well as how NMNAT specifically recognizes pTau and serves as a co-chaperone of Hsp90 for pTau clearance. Our work suggests an interplay of NAD$^+$ metabolism and the progression of Tau pathology in aging and neurodegeneration.

## Results

### The NMNAT family exhibits a conserved chaperone-like activity in preventing pTau aggregation

In the preparation of pTau proteins, we used MARK2 to phosphorylate Tau23 and a truncated construct—K19 (*Figure 1A*; *Gu et al., 2013*). The MARK2 phosphorylation sites on Tau23 and K19 were characterized by 2D $^1$H-$^{15}$N NMR HSQC spectra (*Figure 1A*; *Figure 1—figure supplement 1*). Consistent with previous reports (*Schwalbe et al., 2013*; *Timm et al., 2003*), S262 of repeat region 1 (R1), S324 of repeat region 3 (R3), S352 and S356 of repeat region 4 (R4) were phosphorylated in

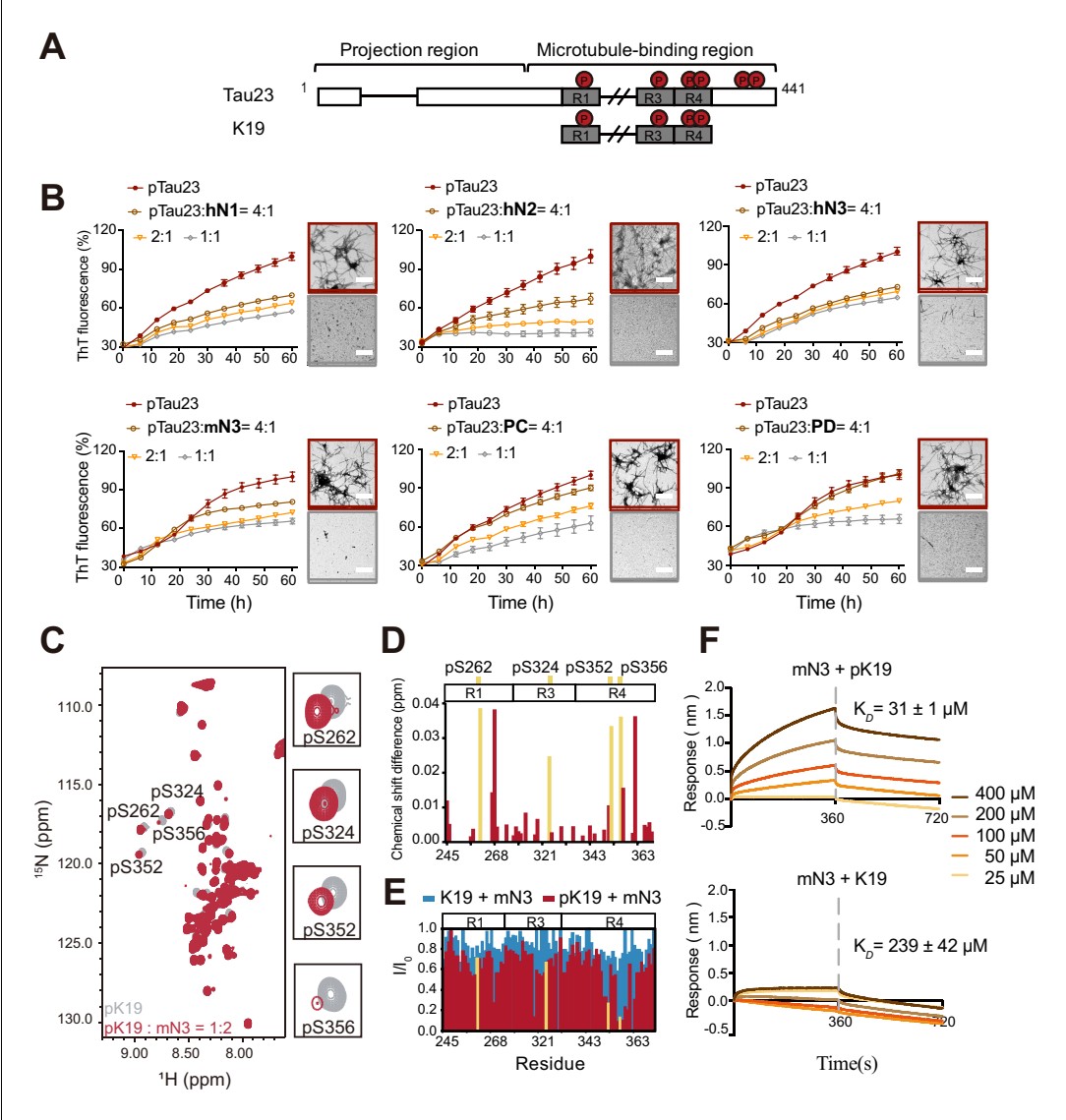

**Figure 1.** NMNATs inhibit pTau amyloid fibril formation and specifically bind to the phosphorylated sites of pTau. (A) Domain compositions of Tau23 and K19. The repeat regions are presented as gray boxes. Phosphorylation sites characterized in this work are marked. (B) Inhibition of NMNATs on the amyloid fibril formation of pTau23 (60 μM) by the ThT fluorescence kinetic assay and TEM microscopy. Human NMNATs: hN1, hN2, and hN3; *Drosophila* Nmnats: PC and PD; mouse NMNAT3: mN3. A gradient concentration of NMNATs were applied as indicated. The data showned correspond to mean ± s.d., with n = 5. The EM images framed in red mean pTau23 only and those in gray mean 1:1 (mol: mol, pTau23:NMNAT). Scale bar: 500 nm. (C) Overlay of 2D $^{1}$H-$^{15}$N HSQC spectra of pK19 alone (100 μM, gray) and pK19 titrated by mN3 (200 μM, red). Signals of pSer residues are enlarged. (D) Residue-specific chemical shift changes of pK19 analyzed based on (c). The domain organization of pK19 is indicated and signals of pSer residues are highlighted in yellow. (E) Residue-specific intensity changes of pK19 (red) and K19 signals (light blue) based on (c) and *Figure 1—figure supplement 6a*. (F) Binding affinity of pK19/K19 with mN3 measured by BLI. The association and dissociation profiles of pK19/K19 to mN3 (20 μg ml$^{-1}$) divided by a vertical dash line are shown. Concentrations of pK19/K19 proteins and dissociation constant ($K_D$) are indicated.

The online version of this article includes the following source data and figure supplement(s) for figure 1:

**Source data 1.** ThT fluorescence profiles of pTau23 in the absence or presence of various NMNATs at the indicated concentrations (*Figure 1B*).

**Source data 2.** Residue-specific chemical shift changes (*Figure 1D*) and intensity changes (*Figure 1E*) of pK19 titrated by mN3.

**Source data 3.** The association and dissociation response of pK19/K19 to mN3 (*Figure 1F*).

**Figure supplement 1.** Characterization of MARK2 phosphorylation sites on pK19 (left) and pTau23 (right) by NMR.

**Figure supplement 2.** Enzyme activities of different isoforms of NMNAT from different organisms.

**Figure supplement 3.** Inhibition of NMNATs on the fibril formation of pTau23 imaged by negative-staining TEM.

**Figure supplement 4.** Inhibition of NMNATs on the fibril formation of pK19.

**Figure supplement 5.** Sequence alignment of NMNATs from different species.

*Figure 1 continued on next page*

*Figure 1 continued*

**Figure supplement 6.** NMR titration of NMNAT to Tau.
**Figure supplement 7.** Binding affinity of pTau23 (left) and Tau23 (right) with mN3 measured by BLI.

both MARK2-treated Tau23 and K19. Besides, pTau23 exhibited two additional phosphorylated sites: S413 and S416.

To comprehensively investigate the activity of NMNAT on pTau aggregation, we prepared different isoforms of NMNAT proteins from different organisms, including three isoforms of human NMNATs (hN1, hN2, and hN3), two isoforms of *Drosophila* NMNATs (a cytoplasmic isoform PD and a nuclear isoform PC [*Ruan et al., 2015*]) and mouse NMNAT3 (mN3). We first confirmed that the NMNAT proteins contained normal enzyme activities (*Figure 1—figure supplement 2*). Then, we conducted the ThT fluorescence kinetic assay and transmission electron microscopy (TEM) to monitor their influences on the amyloid aggregation of pTau. The result showed that different NMNAT isoforms generally exhibited potent chaperone-like activity against the amyloid aggregation of both pTau23 (*Figure 1B*; *Figure 1—source data 1* and *Figure 1—figure supplement 3*) and pK19 (*Figure 1—figure supplement 4A,B*) in a dose-dependent manner. Moreover, no disaggregase activity was observed by the addition of mN3 to preformed pK19 aggregation (*Figure 1—figure supplement 4C*). The results demonstrate that the chaperone-like activity is highly conserved in the NMNAT family across different organisms, which indicates an important biological role of this activity in the protection of pTau from amyloid aggregation.

## Mechanism of the interaction between mN3 and pTau

We next sought to investigate the structural basis of the interaction between NMNAT and pTau. Although hN2 represents the most biological relevant isoform, purified hN2 appeared unstable and prone to aggregate in vitro, which hinders us for further structural characterization. Alternatively, we used mN3, which shares high sequence similarity to hN2 (*Figure 1—figure supplement 5*) and is more stable. We performed solution NMR spectroscopy and used mN3 to titrate $^{15}$N-labeled pK19. The 2D $^1$H-$^{15}$N HSQC spectra showed a significant overall signal broadening of pK19, which indicates a strong interaction between mN3 and pK19 (*Figure 1C*). In particular, the four phosphorylated Ser (pSer) residues showed large signal attenuations and chemical shift changes upon mN3 titration (*Figure 1D,E*; *Figure 1—source data 2*). Residues adjacent to pSer, including regions around a.a. 250, a.a. 320 and a.a. 350, also exhibited prominent signal attenuations (*Figure 1E*; *Figure 1—source data 2*). Especially, repeat sequence R4 that contains two pSer residues showed the largest signal attenuations with $I/I_0 < 0.3$ (*Figure 1E*; *Figure 1—source data 2*). In contrast, as we titrated non-phosphorylated K19 at the same ratio, only slight overall signal broadening was observed in the three regions (*Figure 1E*; *Figure 1—source data 2* and *Figure 1—figure supplement 6A*). Moreover, pTau23, but not Tau23, showed significant chemical shift changes and intensity attenuations mainly on and around the pSer residues upon mN3 titration (*Figure 1—figure supplement 6B*). These results indicate that the pSer residues of pTau are the primary binding sites of mN3.

Further, to quantitatively measure the binding affinity between NMNAT and pTau, we conducted BioLayer Interferometry (BLI) analysis that is a label-free technology for measuring biomolecular interactions (*Rich and Myszka, 2007*). We immobilized mN3 on the biosensor tip and profiled the association and dissociation curves in the presence of either pTau or Tau (*Figure 1F*; *Figure 1—source data 3* and *Figure 1—figure supplement 7*). As we measured, the binding affinity of mN3 to pTau is about one order of magnitude higher than that to Tau. The dissociation constant ($K_D$) for mN3 to pK19 is ~31 μM and to pTau23 is ~9.9 μM. In contrast, the $K_D$ for mN3 to K19 is ~239 μM and to Tau23 is ~58.6 μM (*Figure 1F*; *Figure 1—source data 3* and *Figure 1—figure supplement 7*).

Taken together, these results indicate that MARK2-phosphorylation significantly enhances the interaction between NMNAT and pTau through the specific interaction between NMNAT and the phosphorylated residues of pTau.

## mN3 utilizes its enzymatic substrate-binding site to bind pTau

To identify the binding site of mN3 for pTau, we firstly determined the atomic structure of mN3 at the resolution of 2.0 Å by X-ray crystallography (*Supplementary file 1*). The structure of mN3 monomer is similar to that of hN3 (*Zhang et al., 2003*) with an r.m.s.d. value of 0.543 Å between Cα atoms (*Figure 2—figure supplement 1A*). The catalytic pocket that synthesizes $NAD^+$ from NMN and ATP is highly conserved in the NMNAT family (*Figure 1—figure supplement 5*; *Figure 2—figure supplement 1A*).

The crystal contains two mN3 molecules forming a homo-dimer with a buried surface area of 1,075.8 $Å^2$ in the asymmetric unit (*Figure 2—figure supplement 1B*). Consistently, size exclusion chromatography and multi-angle laser light scattering (SEC-MALS) characterized that mN3 forms a dimer (~64 kDa) in solution (*Figure 2—figure supplement 1C*), and as the mN3 concentration decreased, no significant dissociation of the dimer was observed (*Figure 2—figure supplement 1D*). A similar dimer interface is conserved in *Bacillus subtilis* NMNAT (BsN) (*Olland et al., 2002*), hN1 (*Zhou et al., 2002*), and hN3 (*Zhang et al., 2003*; *Figure 2—figure supplement 1B*). Note that BsN also exists as a dimer in solution (*Olland et al., 2002*). Although hN1 and hN3 exist as tetramer and hexamer in solution, respectively, an equilibrium between dimer and hexamer has been observed (*Zhou et al., 2002*). These indicate that dimer is a functional unit of NMNAT proteins.

To validate that dimerization is required for the function of mN3, we constructed a double mutation of E198P and L217R (referred to as mutation EL) to disrupt the dimer interface. The SEC-MALS result showed that mN3 EL represents a mixture of dimer and monomer in solution (*Figure 2—figure supplement 1E*). The partial dissociation of dimer significantly weakened the enzymatic activity (*Figure 2—figure supplement 1G*), while showed no apparent impact on the chaperone-like activity (*Figure 2—figure supplement 1H*). In addition, the EL mutant is less stable than the WT (*Figure 2—figure supplement 1F*). These results indicate that the dimerization is important for the stability and enzymatic activity of mN3, but less required for the chaperone-like activity. This difference indicates different mechanisms of the dual activities of mN3.

To investigate the mechanism of the chaperone-like activity of mN3, we conducted a cross-linking mass spectrometry (CL-MS) with chemical cross-linker $BS^3$ to covalently link paired lysine residues in spatial proximity (Cα-Cα distance <24 Å) as pTau and mN3 interact, and then identified the cross-linked segments by mass spectrometry. The result showed 7 pairs of cross-linked segments between pK19 and mN3 with a confidence score of $<10^{-7}$ (*Supplementary file 2*). Lysine residues, K95, K139, and K206, that are involved in the cross-linking of mN3 with pK19, cluster around the entrance of the enzymatic pocket of mN3 (*Figure 2A*). The entrance of the enzymatic pocket features a positively-charged patch mainly composed of residues K55, K56, R205 and K206 for the NMN and ATP binding (*Figure 2B*; *Figure 2—figure supplement 1A*). This result implies that mN3 may utilize the same positively charged binding pocket for both pTau and the enzymatic substrates.

To further validate the pTau binding site on mN3, we constructed a quadruple mutation of K55E, K56E, R205E and K206E (referred to as mutation KKRK) to disrupt the positively charged interface. Differential scanning fluorimetry (DSF) confirmed that the mutations did not impair the overall structural stability of mN3 (*Figure 2—figure supplement 2*). To test whether the mutations influence the interaction between mN3 and pTau, we titrated pK19 with the KKRK mutant. The HSQC spectrum showed that the KKRK mutations significantly diminished the affinity of mN3 to the three regions around a.a. 250, a.a. 320 and a.a. 350 that contain pSer residues (*Figure 2C*; *Figure 2—source data 1* and *Figure 2—figure supplement 3*). Especially, the region around a.a. 350 (residues 349–360) of R4, which contains two pSer residues, exhibited a dramatically weakened binding to the KKRK mutant (*Figure 2C*; *Figure 2—source data 1* and *Figure 2—figure supplement 3*).

Furthermore, the KKRK mutations significantly impaired the chaperone-like activity of mN3 against the amyloid aggregation of both pK19 (*Figure 2D*; *Figure 2—source data 2*) and pTau23 (*Figure 2—figure supplement 4A*). Note that the disruption of the positively charged patch did not completely eliminate the chaperone-like activity of mN3, indicating that other interactions also contribute to the binding of NMNAT to pTau. Of note, there is a hydrophobic area on the periphery of the positive-charge patch (*Figure 2B*), implying that hydrophobic interactions may also contribute to the chaperone-like activity of MNNAT to pTau.

Since our NMR data show that segment $^{349}$RVQ(p)SKIG(p)SLDNI$^{360}$, that is shared by both pK19 and pTau23, is a primary binding segment of mN3, we built a complex structure of mN3 and $^{349}$RVQ

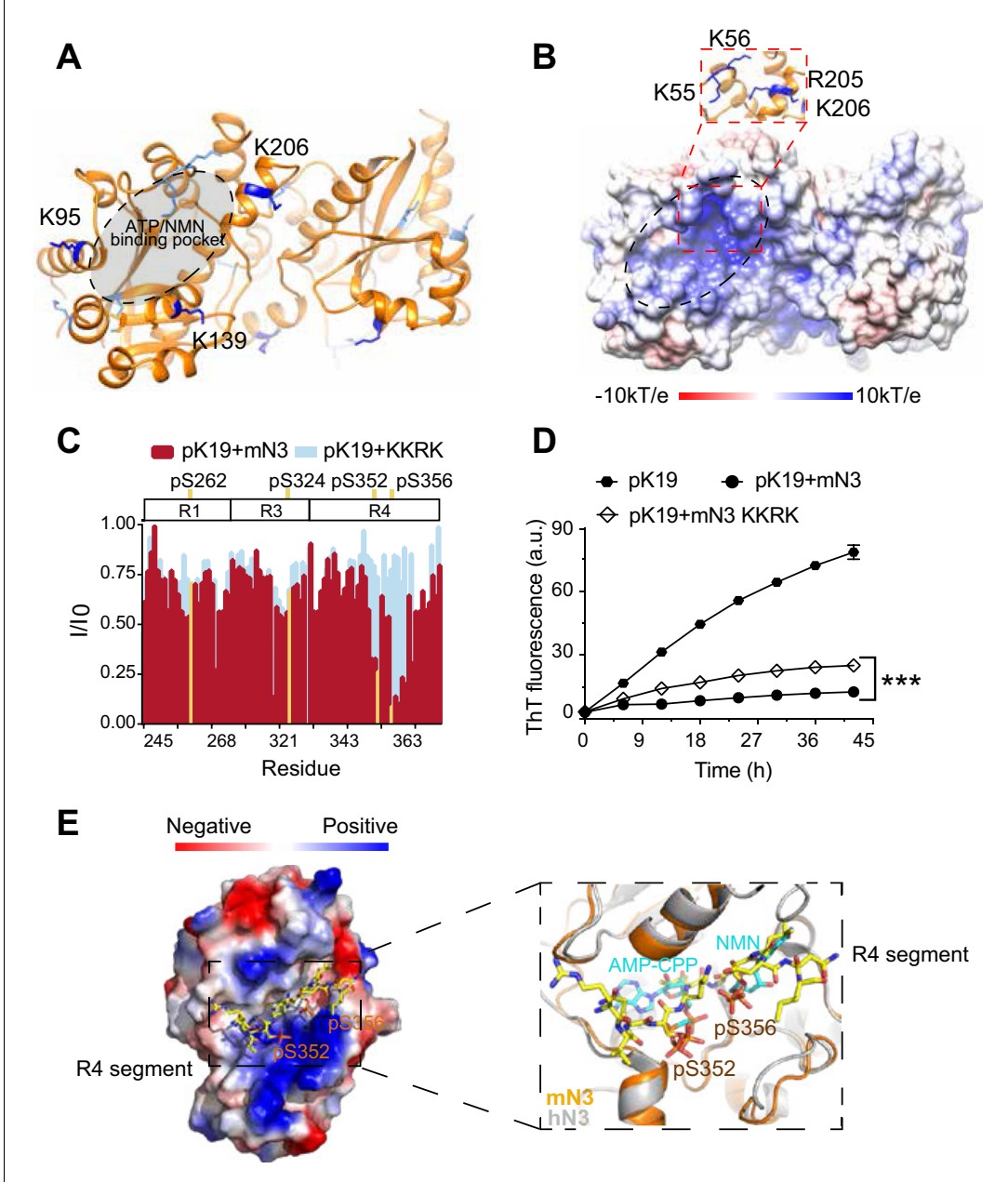

**Figure 2.** Structural characterization of the pTau-binding site on mN3. (**A**) The structure of mN3 is shown in cartoon. Lysine residues that cross-linked with pK19 are shown as sticks in dark blue. Other lysine residues of mN3 are shown in light blue. The ATP/NMN binding site is shaded in gray. (**B**) Electrostatic surface representation of mN3. The *pqr* file was calculated at pH = 6.5. The positively-charged patch is highlighted with dash lines. The residues that compose the positively charged patch are shown as sticks in blue in the zoom-in view. (**C**) Overlay of the residue-specific intensity changes of pK19 signals titrated by mN3 (red) and KKRK mutant (light blue), respectively. pSer residues are colored in yellow. The domain organization of K19 is indicated on top. (**D**) Influence of KKRK mutations on the inhibition of mN3 against pK19 amyloid aggregation measured by the ThT fluorescence assays. The molar ratio of pK19 to mN3 is 1:0.2. Data correspond to mean ± s.d., with n = 3. Values are compared using Student's *t*-test. ****p<0.001*. (**E**) Structural model of mN3 in complex with the phosphorylated R4 segment [349]RVQ(p)SKIG(p)SLDNI[360]. The electrostatic surface of the mN3 structure is shown. The peptide is shown as sticks in yellow. A zoom-in view of the peptide-binding site in (**E**) superimposed on the structure of hN3 in complex with AMP-CPP and NMN (PDB ID: 1NUS). AMP-CPP and NMN are shown as sticks in cyan.

The online version of this article includes the following source data and figure supplement(s) for figure 2:

**Source data 1.** Residue-specific intensity changes of pK19 signals titrated by mN3 KKRK mutant (*Figure 2C*).
**Source data 2.** ThT fluorescence profiles of pK19 in the absence or presence of mN3 WT and KKRK mutant (*Figure 2D*).
**Figure supplement 1.** Characterization of mN3 dimer structure and activities.
**Figure supplement 2.** Melting temperature of mN3 and hN2 variants measured by the DSF assay.

*Figure 2 continued on next page*

*Figure 2 continued*

**Figure supplement 3.** 2D $^1$H-$^{15}$N HSQC spectra of pK19 titrated by mN3-KKRK.

**Figure supplement 4.** Influences of mN3 (**A**) and hN2 (**B**) mutations on pTau23 aggregation measured by ThT fluorescence assays.

(p)SKIG(p)SLDNI$^{360}$ by Rosetta modeling (*Figure 2E*). The complex structural model showed that the phosphorylated segment is well accommodated in the ATP/NMN-binding pocket of mN3. The phosphate groups of pS352 and pS356 orient toward the positively charged pocket of mN3 and position in the same binding site of ATP and NMN (*Figure 2E*). This structure model explains the observation that the KKRK mutant, which impairs the binding of phosphate groups, specifically abolished the binding of mN3 to segment $^{349}$RVQ(p)SKIG(p)SLDNI$^{360}$ of R4 (*Figure 2C*; *Figure 2—source data 1*). In addition, the chemical shift perturbations of pS262 of R1 and pS324 of R3 were also abolished as titrated by the KKRK mutant of mN3 (*Figure 2—figure supplement 3*), which indicates that these pSer residues may bind to mN3 in a similar manner as those of R4.

Primary sequence alignment shows that the key positively charged residues identified for pTau binding are highly conserved in the family of NMNATs from different species (*Figure 1—figure supplement 5*), which suggests that different NMNAT proteins employ a common and conserved interface for pTau binding. Indeed, mutation of the conserved K57 and R274 residues in hN2 severely impaired its chaperone-like activity against pTau23 aggregation (*Figure 2—figure supplement 4B*).

Taken together, these results indicate that NMNAT adopts a conserved pocket to bind both the enzymatic substrates and pTau. Thus, the binding of NMNAT to pTau is similar to the specific binding of enzyme and substrate.

## Competition between pTau and ATP/NMN for NMNAT binding

We next sought to understand how NMNAT spatially organizes its dual functions with the same pocket in a single domain. We have shown that the KKRK mutations of mN3 diminished the chaperone-like activity of NMNAT since they disrupt the positively charged pocket for the binding of phosphate groups. Next, we tested the influence of the KKRK mutations on the enzymatic activity of mN3 on NAD$^+$ synthesis. The result showed that the KKRK mutations also eliminated the enzymatic activity of mN3, which is conceivable due to the inefficient binding of the mutant to the phosphate groups of ATP and NMN (*Figure 3A*; *Figure 3—source data 1*). On the other hand, we mutated H22, a key catalytic residue for NAD$^+$ synthesis (*Saridakis et al., 2001*) that positions deep at the bottom of the substrate-binding pocket (*Figure 3—figure supplement 1A*). The result showed that the H22A mutation resulted in elimination of the enzymatic activity (*Figure 3A*; *Figure 3—source data 1*), which agrees with the previous study on the enzyme activity of NMNAT (*Zhai et al., 2006*). However, we found that the H22A mutation showed no influence on the chaperone-like activity of mN3 in inhibiting the amyloid aggregation of pK19 (*Figure 3B*; *Figure 3—source data 2*).

To examine the competition of the two activities, we used the BLI analysis and found that as the concentration of NMN increased, the binding of pK19 to mN3 was remarkably weakened (*Figure 3C*; *Figure 3—source data 3*). The EC50 of NMN as a competitor for mN3's interaction with pTau is 501 µM. Consistently, the ThT assays showed that as the concentrations NMN or ATP decreased, the chaperone-like activity of mN3 on the amyloid aggregation of pK19 dramatically increased (*Figure 3D*; *Figure 3—source data 4*). In contrast, as we reversely added pK19 into the enzymatic reaction of NAD$^+$ synthesis, no significant influence was observed (*Figure 3E*; *Figure 3—source data 5*).

Taken together, our data indicate that the enzymatic substrates (i.e. NMN and ATP) and the chaperone client pTau of mN3 share the same binding pocket with a partial overlap at the phosphate-binding site, while ATP and NMN are superior to pTau on the mN3 binding.

## Nmnat protects pTau-induced synaptopathy in *Drosophila*

To assess the functional relevance of the direct regulation of NMNAT on the abnormal aggregation of pTau in vivo, we examined the protective capability of Nmnat in *Drosophila* tauopathy models by overexpressing human wild type (Tau$^{WT}$) or pathogenic Tau (Tau$^{R406W}$) in the visual system using a photoreceptor-specific driver *GMR-GAL4* (*Ali et al., 2012*). The expression pattern can be easily visualized due to the highly organized parallel structure of the compound eye: the R1-R6

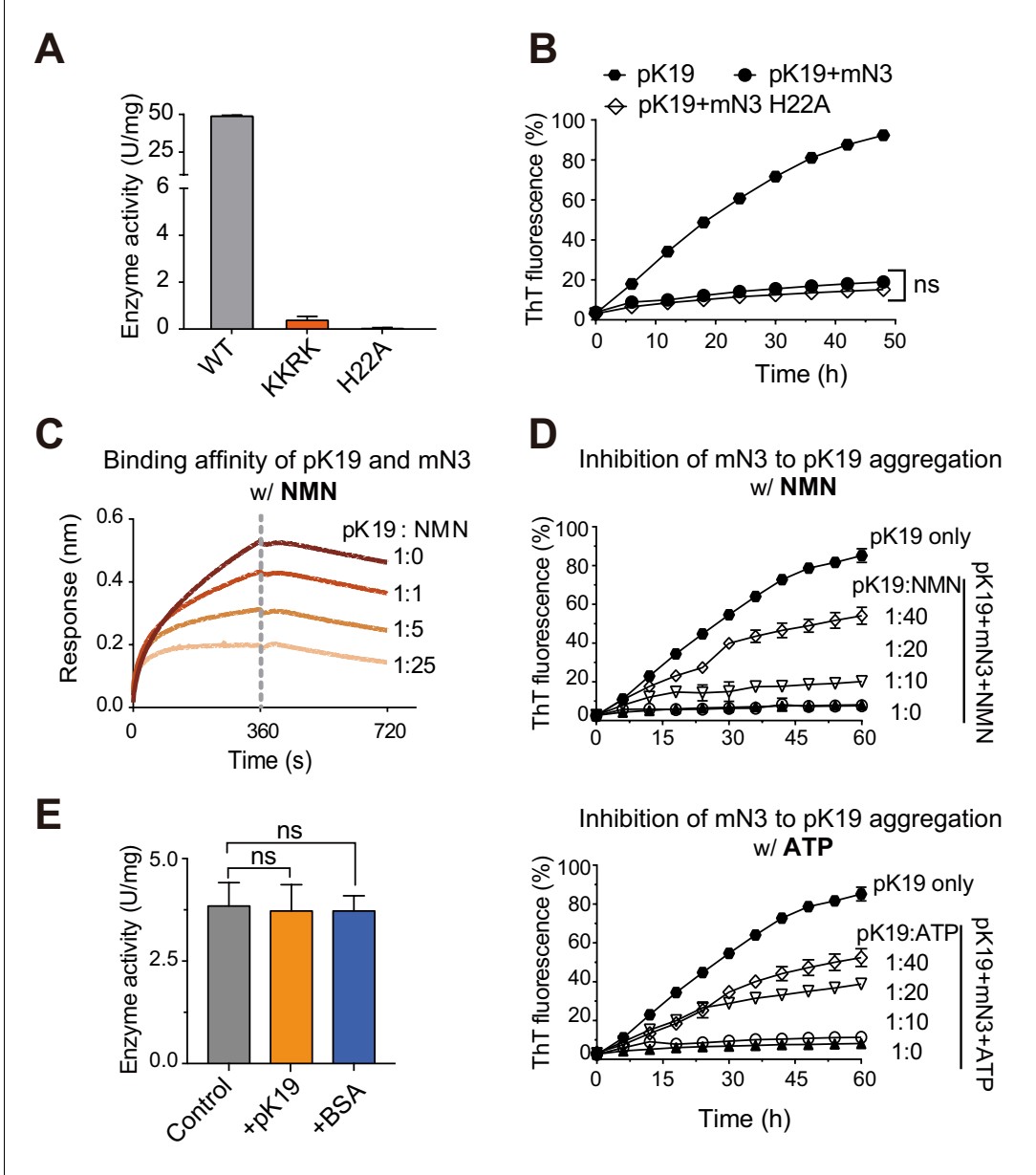

**Figure 3.** Competition of pK19 and NMN/ATP in the dual activities of NMNAT. (**A**) Enzyme activities of mN3 WT and mutants. The data showed correspond to mean ± s.d., with n = 3. (**B**) Influence of the H22A mutation on the inhibition of mN3 against the amyloid aggregation of pK19. The molar ratio of pK19 to mN3 is 1:0.2. The data showed correspond to mean ± s.d., with n = 3. Values are compared using Student's *t*-test. 'ns', not significant. (**C**) NMN weakens the binding of pK19 to mN3 in a dose-dependent manner measured by BLI analysis. mN3 was immobilized on the SA sensor. pK19 (50 μM) was pre-mixed with NMN at the indicated molar ratios for association. The same amounts of NMN were used for the association and dissociation measurements. The association and dissociation profiles are divided by a vertical dash line. (**D**) The presence of NMN (top) or ATP (bottom) reduces the inhibitory effect of mN3 against pK19 amyloid aggregation in a dose-dependent manner. Molar ratios of pK19 to NMN/ATP are indicated. The data showed correspond to mean ± s.d., with n = 3. (**E**) The presence of pK19 (pK19: NMN/ATP = 10:1) shows no significant influence on the enzyme activity of mN3. The data showed correspond to mean ± s.d., with n = 3. Values are compared using Student's *t*-test. 'ns', not significant. The online version of this article includes the following source data and figure supplement(s) for figure 3:

**Source data 1.** Enzyme activities of mN3 WT and mutants (*Figure 3A*).
**Source data 2.** ThT fluorescence profiles of pK19 in the absence or presence of mN3 WT and H22A mutant (*Figure 3B*).
**Source data 3.** NMN weakens the binding of pK19 to mN3 (*Figure 3C*).
**Source data 4.** The presence of NMN or ATP reduces the inhibitory effect of mN3 against pK19 amyloid aggregation (*Figure 3D*).
**Source data 5.** The presence of pK19 shows no significant influence on the enzyme activity of mN3 (*Figure 3E*).
**Figure supplement 1.** Position of H22 in the over-all mN3 structure.

photoreceptors have their axons traverse the lamina cortex (*Figure 4A,B*, magenta box) and make synaptic contacts at the lamina neuropil (*Figure 4A,B*, red box), while R7-R8 photoreceptors extend their axons beyond lamina and project to distinct layers in medullar neuropil (*Figure 4A,B*, orange box) (*Sato et al., 2013*). We found that both pTau$^{WT}$ and pTau$^{R406W}$ aggregated in the brain, which could be suppressed by PD overexpression (*Figure 4A*; *Figure 4—figure supplement 1A*, white arrowheads). Compared to Tau$^{WT}$, Tau$^{R406W}$ exhibited a more severe retinal degeneration in the

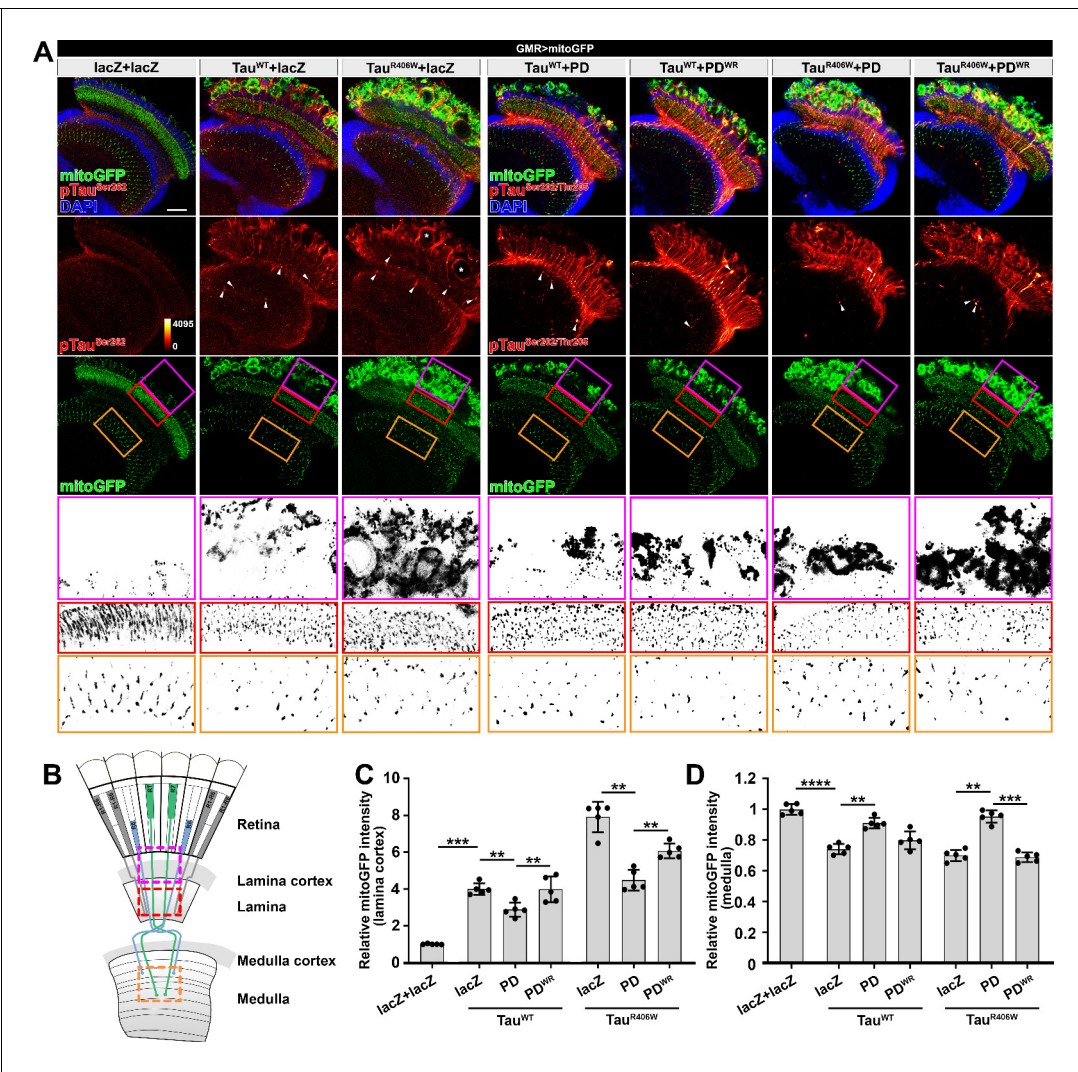

**Figure 4.** *Drosophila* Nmnat (PD) suppresses pTau-induced mitochondrial clustering. (**A**) Adult female *Drosophila* (2 days after eclosion, DAE) brains expressing mitochondrial marker mitoGFP (green) together with lacZ+lacZ, Tau$^{WT}$+lacZ, Tau$^{R406W}$+lacZ, Tau$^{WT}$+PD, Tau$^{WT}$+PD$^{WR}$, Tau$^{R406W}$+PD, or Tau$^{R406W}$+PD$^{WR}$ under photoreceptor-specific driver *GMR-GAL4* were stained for pTau (red spectrum) and DAPI (blue). White arrowheads show the aggregation of pTau. White asterisks show the holes formed in the lamina cortex layer, indicating retinal degeneration. Magenta, red, and yellow boxes indicate the lamina cortex, lamina, and medulla layers, respectively. Scale bar, 30 μm. (**B**) Diagram of the adult *Drosophila* visual system. Each ommatidium contains six outer photoreceptors (R1–R6) and two inner photoreceptors (R7 and R8). R1-R6 traverse the lamina cortex (magenta box) and project their axons into the lamina (red box), while the axons of R7 and R8 pass through the lamina and terminate in distinct synaptic layers in the medulla (orange box). (**C, D**) Quantification of mitoGFP intensity in the lamina cortex (**C**) and medulla (**D**). Data are presented as mean ± s.d., with n = 5. One-way ANOVA post hoc Tukey test; \*\*p<0.01, \*\*\*p<0.001, \*\*\*\*p<0.0001.

The online version of this article includes the following figure supplement(s) for figure 4:

**Figure supplement 1.** *Drosophila* NMNAT(PD) suppresses pTau-induced mitochondrial clustering.

**Figure supplement 2.** *Drosophila* Nmnat (PD) suppresses pTau-induced Brp loss.

**Figure supplement 3.** Activities of mN3 variants.

**Figure supplement 4.** *Drosophila* Nmnat PD and PD$^{WR}$ colocalize with pTau in vivo.

lamina cortex (*Figure 4A*; *Figure 4—figure supplement 1A*, white asterisks), which was mitigated by PD overexpression. Hyperphosphorylated and aggregated Tau can impair axonal transport, leading to abnormal mitochondrial distribution and clustering (*Ebneth et al., 1998*). Consistently, we found that overexpression of Tau species, especially Tau$^{R406W}$, led to remarkable clustering of mitochondria in the lamina cortex, fragmentation of mitochondria in the lamina, and loss of mitochondria at R7-R8 terminals (*Figure 4*; *Figure 4—figure supplement 1A*). Overexpression of PD significantly suppressed mitochondrial clustering in the lamina cortex and restored mitochondria localization at R7 and R8 terminals.

Synaptic loss has been characterized as one of the main lesions presented in human tauopathies (*McGowan et al., 2006*). To evaluate the synaptic integrity, we next examined the distribution of Bruchpilot (Brp), an active zone associated cytoskeletal matrix protein. In wildtype lamina, Brp staining of the cross-sections reveals a pattern consists of repetitive cartridge structure, which is formed by R1-R6 photoreceptor terminals (*Figure 4—figure supplement 2A*). Tau$^{R406W}$ overexpression resulted in synaptic aggregation of hyperphosphorylated Tau (*Figure 4—figure supplement 2A*, arrowheads) and ~50% reduction of the Brp levels within each lamina cartridge compared with that in the wild type flies, suggesting a severe loss of the active zone structures in the presynaptic terminals. Remarkably, the pTau aggregation and the synaptic phenotype can be suppressed by overexpressing PD, as suggested by significantly restored endogenous Brp levels (*Figure 4—figure supplement 2*).

It was previously described that Tau increases F-actin aggregation at synaptic terminals, promotes actin polymerization, and reduces synaptic vesicle mobility (*Zhou et al., 2017*). Consistently, we found an increased level of F-actin at R7 and R8 terminals with Tau$^{WT}$ or Tau$^{R406W}$ expression, which could be suppressed by PD overexpression (*Figure 5A and B*).

To dissect the protective mechanism of PD, we used a *Drosophila* line with *UAS-PD$^{WR}$* (W98G/R224A double mutant) insertion that can be used to express an enzyme-inactive PD. The R224 residue in *Drosophila* is corresponding to the R205 residue in mN3. Our in vitro data show that PD$^{WR}$ has abolished enzymatic activity and modestly decreased chaperone-like activity (*Figure 4—figure supplement 3*). In the *Drosophila* tauopathy model, compared to wild type PD, PD$^{WR}$ shows modestly increased mitoGFP clustering at the lamina cortex, reduced mitoGFP and increased F-actin accumulation at synaptic terminals, indicating that PD$^{WR}$ has reduced protective capacity in vivo. Notably, we did not find a significant difference in pTau level when overexpressing wild type PD and PD$^{WR}$ (*Figure 4—figure supplement 1*), suggesting that the decreased protective capacity of PD$^{WR}$ is not due to an alteration of pTau clearance capability. To further examine the interaction between PD and pTau in vivo, we co-expressed Tau$^{R406W}$ and PD/PD$^{WR}$ in the third-instar larval salivary gland cells to take advantage of their larger size and better spatial resolution. We found that both wild type PD and PD$^{WR}$ colocalize with pTau (*Figure 4—figure supplement 4*), indicating a direct interaction between PD/PD$^{WR}$ and pTau.

Of note, although both *Drosophila* Nmnat isoforms PC and PD exhibited potent chaperone-like activity against pTau aggregation in vitro (*Figure 1—figure supplement 3*; *Figure 1—figure supplement 4*), they exhibit distinct protective capacities in vivo. While PD shows potent protection against tauopathy as evidenced by reduced pTau level, reduced brain apoptosis, and improved locomotor activity, PC has a minimal protective capacity (*Figure 5—figure supplement 1*; *Figure 5—figure supplement 2*). Therefore, the protective capacity of Nmnat in vivo depends on its subcellular localization, as the cytoplasmically localized PD isoform is likely more accessible to the client pTau protein.

Taken together, our data demonstrate that pTau interferes with synaptic functions via (1) impairing mitochondrial dynamics and localization in neurons, (2) disrupting synaptic active zone integrity, and (3) stimulating F-actin accumulation that restricts synaptic vesicle mobility and release. Nmnat specifically binds to pTau and protects against pTau-induced synaptic dysfunction by regulating pTau aggregation, restoring mitochondria and Brp localization at synaptic terminals, and alleviating pathological F-actin accumulation (*Figure 5C*).

## mN3 mediates the recognition of Hsp90 to pTau

Previous studies showed that hN2 and Hsp90 co-precipitate with pTau in the brains of AD patients, and exhibit a synergistic effect on the attenuation of pTau pathology in cell models (*Ali et al., 2016*), Here, we used the single-molecule pull-down (SMPull) assay to identify the interplay between

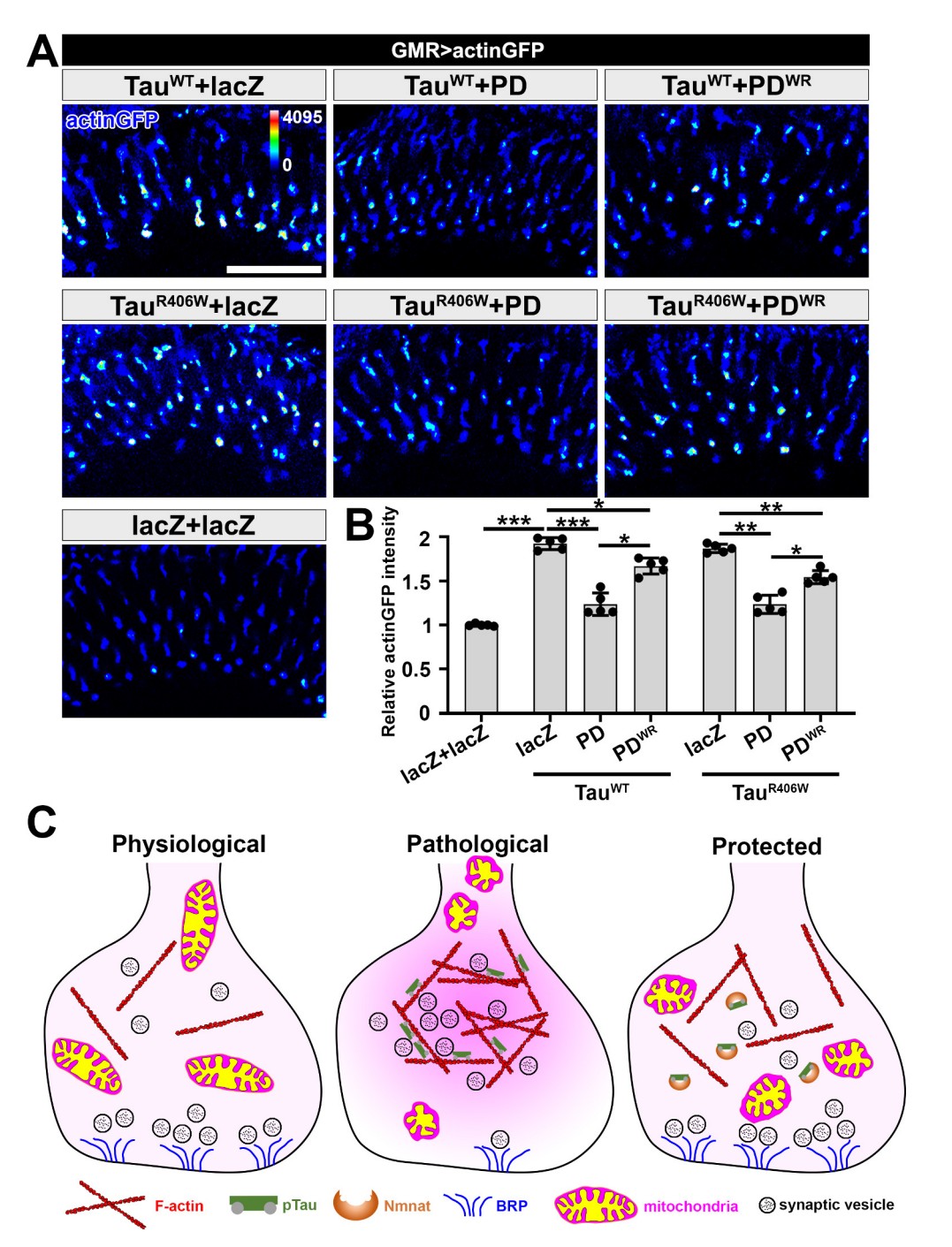

**Figure 5.** NMNAT (PD) suppresses F-actin accumulation at synaptic terminals. (**A**) Medulla area of adult female *Drosophila* (2 DAE) expressing actin-GFP (blue spectrum) together with lacZ+lacZ, Tau$^{WT}$+lacZ, Tau$^{WT}$+PD, Tau$^{WT}$+PD$^{WR}$, Tau$^{R406W}$+lacZ, Tau$^{R406W}$+PD, or Tau$^{R406W}$+PD$^{WR}$ under photoreceptor-specific driver *GMR-GAL4*. Scale bar, 30 µm. (**B**) Quantification of the actinGFP level in the medulla. One-way ANOVA post hoc Tukey test; *$p<0.05$, **$p<0.01$, ***$p<0.001$. (**C**) Schematic model of Nmnat protection against pTau-induced synaptopathy. Under pathological condition, pTau promotes mitochondrial clustering and impairs mitochondrial transport, reduces Brp level at synapses, and stimulates F-actin accumulation that restricts synaptic vesicle mobility. Nmnat specifically binds to pTau and regulates pTau phase separation, restores the presence of mitochondria and Brp at synaptic terminals, and alleviates F-actin accumulation.

The online version of this article includes the following figure supplement(s) for figure 5:

**Figure supplement 1.** *Drosophila* Nmnat-PD, not PC, reduces total pTau$^{Ser262}$ level in the brain.

**Figure supplement 2.** *Drosophila* Nmnat PD, not PC, protects against Tau-induced brain apoptosis and locomotor defects.

NMNAT, Hsp90, and pTau. SMPull is a powerful tool to quantitatively detect weak and transient interactions between protein complexes. As shown in *Figure 6A*, His$_6$-tagged Hsp90 was coated on the slide, and the binding of pTau23 can be detected by the fluorescence from Alexa-647-labeled pTau23 using the total internal reflection fluorescence (TIRF) microscopy. The result showed that in the absence of mN3, binding of pTau23 to Hsp90 was only at the basal level similar to that of the blank slide (~20 fluorescent spots per imaging area), which indicates that the interaction between them is very weak and transient. However, the addition of mN3 to the Hsp90/pTau23 system significantly increased the fluorescent spots in a dose-dependent manner (*Figure 6B*; *Figure 6—figure supplement 1A* and *Figure 6—source data 1*). In contrast, the binding of non-phosphorylated Tau23 to Hsp90 is not affected by the addition of mN3 (*Figure 6—figure supplement 1A,B*; *Figure 6—source data 1*). Thus, these results indicate that mN3 mediates the binding of pTau23 to Hsp90.

Furthermore, The BLI analysis showed that Hsp90 directly bound to mN3 with a $K_D$ value of ~1.93 µM (*Figure 6C*; *Figure 6—source data 2*). However, Hsp90 was not able to differentiate pTau23 from Tau23 with the binding affinity of 16.1 µM to pTau23 and 16.2 µM to Tau23 (*Figure 6D*;

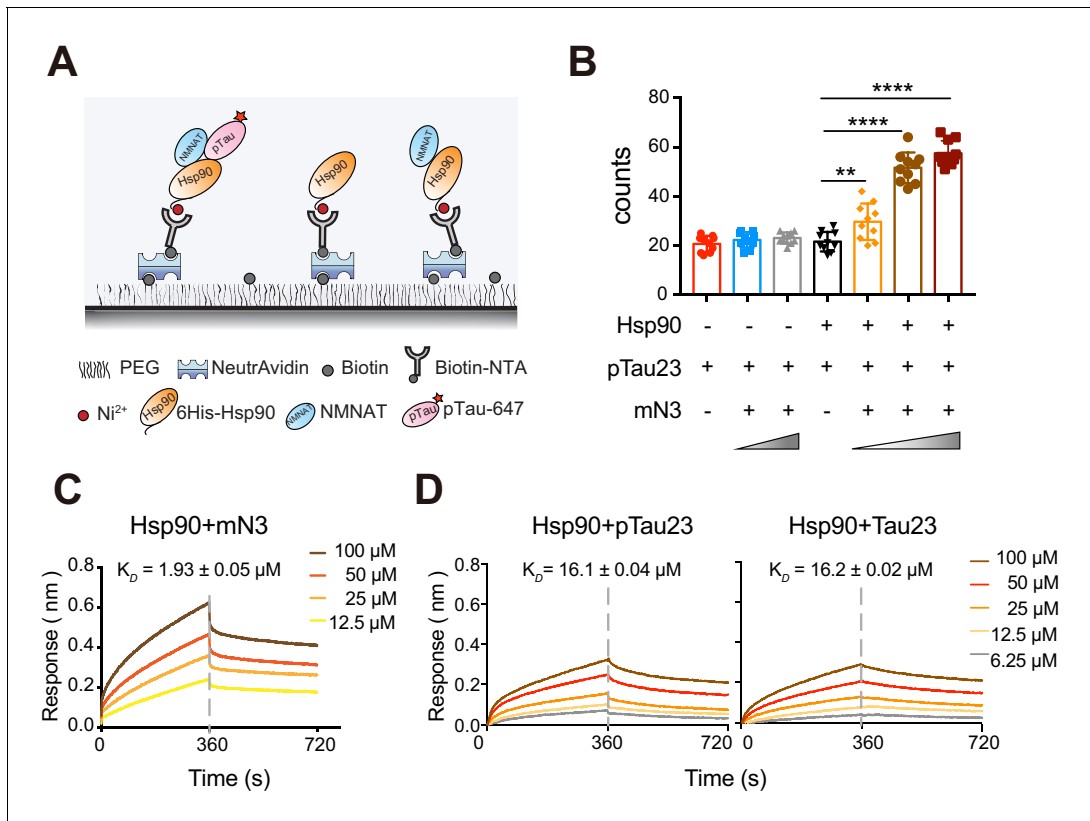

**Figure 6.** mN3 acts as a co-chaperone to assist Hsp90 in the recognition of pTau. (**A**) Schematic illustration of the SMPull assay by TIRF microscopy. His$_6$-tagged Hsp90 was immobilized to the slide by chelating to Biotin-NTA-Ni. Single molecular interaction was monitored by the fluorescence from Alexa-647 that was labeled on pTau23/Tau23 monomer. (**B**) The average number of fluorescent counts per imaging area detected by SMPull. TIRF images were recorded for the sample systems containing Hsp90, mN3 (4 nM) and pTau23 as indicated. The concentrations of mN3 from left to right are 0, 4, 20, 0, 0.8, 4, and 20 nM. Error bars denote standard deviations (s.d.) (n = 10). Values were compared using Student's *t*-test. **, *p<0.01*. ****, *p<0.0001*. (**C**) BLI measurements of mN3 binding to the SA sensor chip coated with biotinylated Hsp90 (20 µg ml$^{-1}$). The mN3 concentrations are indicated. The $K_D$ value of mN3 binding to Hsp90 is reported. (**D**) BLI measurements of the binding of pTau23 (left)/Tau23 (right) to the SA sensor chip coated with biotinylated Hsp90 (20 µg ml$^{-1}$). The Tau protein concentrations are indicated.

The online version of this article includes the following source data and figure supplement(s) for figure 6:

**Source data 1.** The average number of fluorescent counts per imaging area detected by SMPull (*Figure 6B* and *Figure 6—figure supplement 1*).
**Source data 2.** BLI measurements of the binding of mN3 (*Figure 6C*) or pTau23/Tau23 (*Figure 6D*) to Hsp90.
**Figure supplement 1.** mN3 mediates the binding of pTau23, but not Tau23, to Hsp90.

*Figure 6—source data 2*). Taken together, our data indicate that NMNAT acts as a co-chaperone to assist Hsp90 in the recognition of pTau.

## Discussion

### NMNAT is distinct from canonical molecular chaperones

NMNAT proteins have shown a robust neuroprotective activity in various models of neurodegenerative diseases correlated with the decrease of amyloid protein aggregation (*Ali et al., 2013*; *Brazill et al., 2017*; *Conforti et al., 2014*). In this work, we demonstrate that NMNAT functions similar to a molecular chaperone to protect pTau from amyloid aggregation. Our work uncovers the structural basis underlying the binding of NMNAT with pTau and how NMNAT manages its dual enzymatic and chaperone-like functions. As illustrated in *Figure 7*, as NMNAT binds its enzymatic substrates, i.e. ATP and NMN, the substrates settle deep inside the pocket with defined interactions with NMNAT. As for the binding of pTau, the phosphorylated residues of pTau can specifically dock into the phosphate binding sites of NMNAT, which partially overlaps with the binding of ATP and NMN.

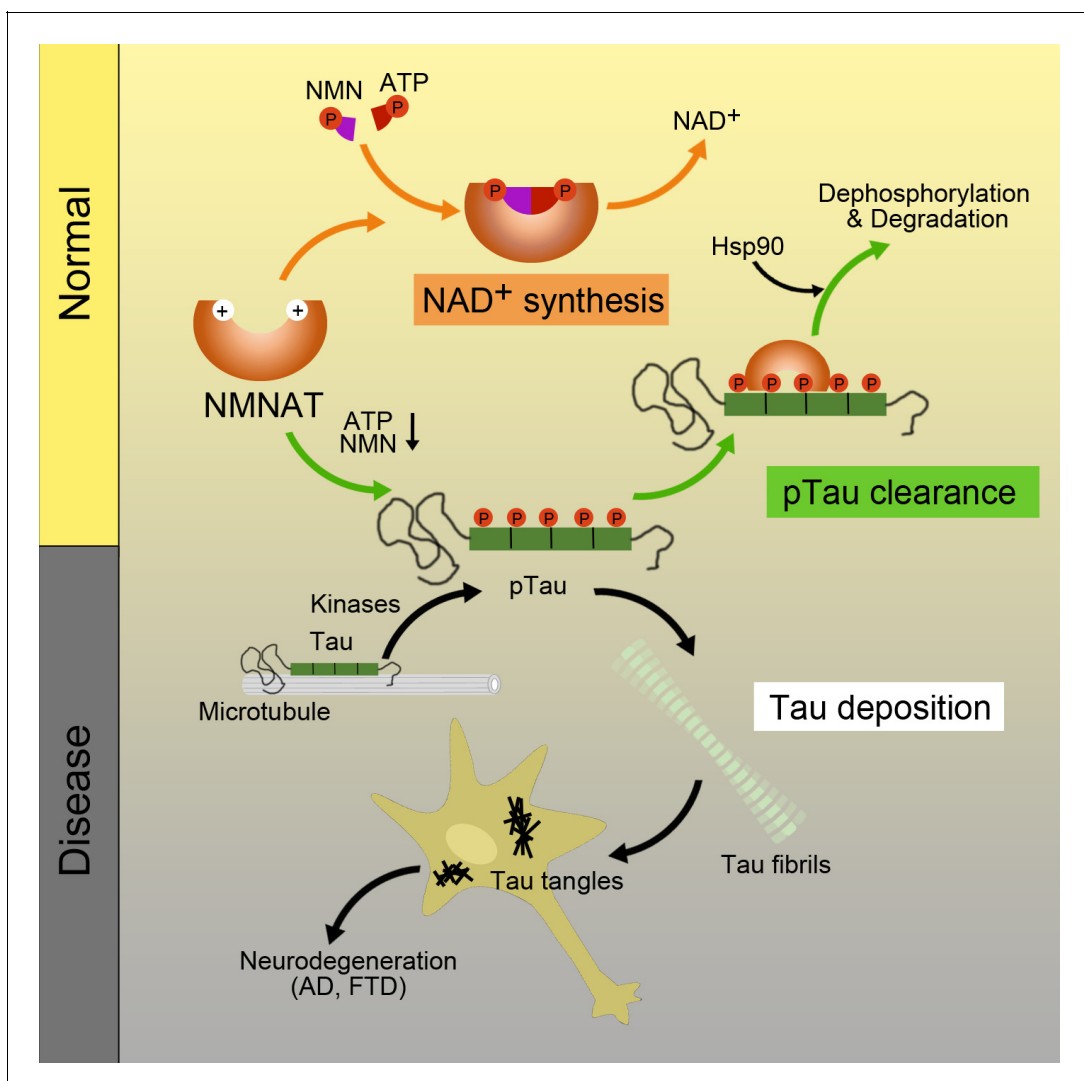

**Figure 7.** Schematics of NMNAT as a key node between pTau homeostasis and NAD$^+$ metabolism. NMNAT functions as both an NAD$^+$ synthase involved in NAD$^+$ metabolism, and a chaperone-like protein assisting the clearance of pathological pTau deposition. During aging, as the levels of ATP and NMN decrease, the chaperone-like function of NMNAT may show up to antagonize pTau aggregation.

It is important to point out that although phosphorylation remarkably increases the binding affinity of mN3 to pTau, without phosphorylation, mN3 still shows weak interactions with the KXGS motifs of Tau (*Figure 1E*; *Figure 1—source data 2* and *Figure 1—figure supplement 6B*). As we stated earlier, there is a hydrophobic area on the periphery of the phosphate-binding surface (*Figure 2B*), indicating the role of hydrophobic interactions in the NMNAT-client binding. Indeed, a recent study reported an interaction between NMNAT and mutant Huntingtin (*Zhu et al., 2019*). Huntingtin is uncharged and not known to be associated with hyperphosphorylation. Therefore, NMNAT-client interaction may be versatile. Our study indicates both electrostatic and hydrophobic interactions are involved in the NMNAT-client binding, while NMNAT specifically binds to pTau through its enzymatic pocket.

NMNAT is distinct from canonical chaperones. Canonical chaperones are either ATP-dependent or ATP-independent. In contrast, although NMNAT is able to bind and hydrolyze ATP, its chaperone-like activity is independent of ATP hydrolysis. NMNAT utilizes the same domain, even a shared pocket, for the binding of ATP and pTau, which explains the confusing previous observation that the neuroprotective effect of NMNAT is independent of ATP hydrolysis, yet requires the integrity of the ATP binding site (*Ali et al., 2016*). In addition, no large conformational change occurs as NMNAT binding to pTau.

In addition to inhibiting pTau aggregation, NMNAT assists Hsp90 in the selection for pTau, implying that NMNAT serves as a co-chaperone of Hsp90 for pTau clearance. Previous studies identified CHIP (carboxyl terminus of the Hsc70-interacting protein) as an important co-chaperone of Hsp90 for pTau removal (*Dickey et al., 2007a*; *Dickey et al., 2007b*). Whereas, CHIP-Hsp90 only recognizes Tau phosphorylated at proline-directed Ser/Thr sites, but not KXGS (MARK) sites (*Dickey et al., 2007a*). Our study shows that NMNAT-Hsp90 can recognize Tau phosphorylated at MARK2-phosphorylated KXGS sites. Therefore, Hsp90 employs different co-chaperones to handle different species of Tau and maintain Tau homeostasis.

## NMNAT links NAD$^+$ metabolism and tau homeostasis

Our study demonstrates that the defined positively-charged pocket for phosphate binding of NMNAT gives rise to the binding of pTau with a high affinity comparable to that of an enzyme-substrate binding, but distinct from the weak interaction commonly found in chaperone-client binding (*Koldewey et al., 2017*). The co-existence of the NAD$^+$ synthetic and chaperone-like activity sharing a common surface of NMNAT indicates a connection between NAD$^+$ metabolism and Tau proteostasis.

NMN is a key intermediate of NAD$^+$. The levels of NMN and NAD$^+$ depend on the activity of nicotinamide phosphoribosyltransferase (NAMPT) that converts nicotinamide and PRPP to NMN and is a rate-limiting enzyme in NAD$^+$ synthesis (*Stein and Imai, 2014*). It has been shown that NAMPT and NAD$^+$ levels decline during aging (*Stein and Imai, 2014*; *Gomes et al., 2013*). Therefore, we may assume that in young and healthy neurons, where the pTau level is low and the NAD$^+$ synthesis is normal, NMNAT would mainly function to generate NAD$^+$. In contrast, in old and degenerating neurons, where pTau increases and NAD$^+$ synthesis declines, NMNAT would switch its function to inhibit pTau aggregation and assist Hsp90 for pTau clearance (*Figure 7*).

The cellular processes of NAD$^+$ metabolism, and Tau phosphorylation and proteostasis are complex with multiple nodes of regulation. NMNAT emerges as a critical regulator balancing between the NAD$^+$-involved active metabolic state and the amyloid-accumulating proteotoxic stress state. Such regulation would be particularly important to maintain the structural and functional integrity of neurons.

## Materials and methods

**Key resources table**

| Reagent type (species) or resource | Designation | Source or reference | Identifiers | Additional information |
|---|---|---|---|---|

*Continued on next page*

*Continued*

| Reagent type (species) or resource | Designation | Source or reference | Identifiers | Additional information |
|---|---|---|---|---|
| Antibody | Rabbit polyclonal anti-cleaved caspase 3 (Asp175) | Cell Signaling Technology | Cat# 9661; RRID:AB_2341188 | (1:1000) for western blot |
| Antibody | Mouse monoclonal anti-β-actin | Sigma-Aldrich | Cat# A1978; RRID:AB_476692 | (1:10000) for western blot |
| Antibody | Mouse monoclonal anti-Brp | Developmental Studies Hybridoma Bank | Cat# nc82; RRID:AB_2314866 | (1:250) for immuno fluorescent staining |
| Antibody | Rabbit polyclonal anti-pTau(Ser262) | Santa Cruz | Cat# sc-101813; RRID:AB_1129981 | (1:250) for immuno fluorescent staining |
| Antibody | Mouse monoclonal anti-pTau (Ser202/Thr205) | Thermo Fisher Scientific | Cat# MN1020; RRID:AB_223647 | (1:250) for immuno fluorescent staining |
| Antibody | Guinea pig polyclonal anti-*Drosophila* Nmnat | Laboratory of Dr. R Grace Zhai **Zhai et al., 2006** | | (1:500) for immuno fluorescent staining |
| Antibody | Alexa Fluor 555 Goat polyclonal anti-Mouse IgG (H+L) | Thermo Fisher Scientific | Cat# A-21422; RRID:AB_2535844 | (1:250) for immuno fluorescent staining |
| Antibody | Alexa Fluor 647 Goat polyclonal anti-Guinea pig IgG (H+L) | Thermo Fisher Scientific | Cat# A-21450; RRID:AB_2735091 | (1:250) for immuno fluorescent staining |
| Antibody | Cy3 conjugated polyclonal anti-Rabbit IgG (H+L) | Rockland | Cat# 611-104-122; RRID:AB_218568 | (1:250) for immuno fluorescent staining |
| Antibody | DyLight 800 conjugated polyclonal anti-Mouse IgG (H+L) | Rockland | Cat# 610-145-002; RRID:AB_10703265 | (1:1000) for western blot |
| Antibody | DyLight 680 conjugated polyclonal anti-Rabbit IgG (H+L) | Rockland | Cat# 611-144-002; RRID:AB_1660962 | (1:1000) for western blot |
| Chemical | Protease inhibitor | Roche | Cat# 11873580001 | |
| Chemical | Triton X-100 | Sigma-Aldrich | Cat# T9284 | |
| Chemical | VECTASHIELD antifade mounting medium | Vector Laboratories | Cat# H-1000; RRID:AB_2336789 | |
| Chemical | DAPI | Thermo Fisher Scientific | Cat# D1306; RRID:AB_2629482 | (1:300) for immuno fluorescent staining |
| Chemical | Normal goat serum | Thermo Fisher Scientific | Cat# PCN5000 | |
| Peptide, recombinant protein | His6-NMNATs | This paper | | purified from *E. coli* BL21-RIL cells |
| Peptide, recombinant protein | His6-Hsp90 | This paper | | purified from *E. coli* BL21-RIL cells |
| Peptide, recombinant protein | Tau23 | This paper | | purified from *E. coli* BL21-RIL cells |
| Peptide, recombinant protein | K19 | This paper | | purified from *E. coli* BL21-RIL cells |
| Peptide, recombinant protein | His10-HRV-3C protease | This paper | | purified from *E. coli* BL21-RIL cells |
| Peptide, recombinant protein | His6-MBP-MARK2-T208E | This paper | | purified from *E. coli* BL21-RIL cells |

*Continued on next page*

*Continued*

| Reagent type (species) or resource | Designation | Source or reference | Identifiers | Additional information |
|---|---|---|---|---|
| Chemical compound, drug | Thioflavin T | Sigma-Aldrich | Cat# 596200 | |
| Chemical compound, drug | BS$^3$ | Thermo Fisher Scientific | Cat# 21585 | |
| Chemical compound, drug | SYPRO Orange | Thermo Fisher Scientific | Cat# S6650 | |
| Chemical compound, drug | NMN | Sigma-Aldrich | Cat# N3501 | |
| Chemical compound, drug | ATP | Sigma-Aldrich | Cat# A2383 | |
| Chemical compound, drug | Semicarbazide-HCl | Sigma-Aldrich | Cat# S2201 | |
| Chemical compound, drug | Alcohol dehydrogenase | Sigma-Aldrich | Cat# A7011 | |
| Chemical compound, drug | Alexa Fluor 647 | Thermo Fisher Scientific | Cat# A32757 | |
| Software, algorithm | Graphpad Prism | GraphPad software | SCR_002798 | |
| Software, algorithm | Adobe illustrator | Adobe Inc | SCR_010279 | |
| Software, algorithm | Adobe Photoshop | Adobe Inc | SCR_014199 | |
| Software, algorithm | ASTRA VI software | Wyatt Technologies | SCR_016255 | |
| Software, algorithm | smCamera | Taekjip Ha, Johns Hopkins University | | |
| Software, algorithm | LI-COR Image Studio Software | LI-COR Biosciences | SCR_015795 | |
| Software, algorithm | Olympus Fluoview FV10-ASW | Olympus | SCR_014215 | |

## Protein expression and purification

Genes encoding mN1, mN3, *Drosophila* PC, PD (gift from Dr. Yanshan Fang) and genes encoding hN1, hN3 (purchased from GENEWIZ, Inc Suzhou, China) were amplified and inserted into pET-28a vector with an N-terminal His$_6$-tag and a following thrombin cleavage site. Gene encoding hN2 (purchased from Genewiz, Inc) was cloned into pET-32M-3C derived from pET-32a (Novagen). The resulting plasmid encodes a protein with an N-terminal MBP (maltose-binding protein) and a His$_6$-tag followed by an HRV 3C protease recognition site. Mutations of mN3 including KK (K55EK56E), RK (R205EK206E), KKRK (K55EK56ER205EK206E) and H22A were constructed by site-directed mutagenesis using Q5 Site-Directed Mutagenesis Kit (New England Biolabs). All the resulting constructs were verified by DNA sequencing (GENEWIZ, Inc Suzhou, China).

NMNATs and variants were over-expressed in *E. coli* BL21 (DE3) cells. Cells were grown 2 × YT medium at 37°C to an OD$_{600}$ of 0.8–1. Protein expression was induced by the addition of 0.2 mM isopropyl-β-d-1-thiogalactopyranoside (IPTG) and incubated at 16°C for 15 hr. Cells were harvested by centrifugation at 4,000 *rpm* for 20 min and lysed in 50 ml lysis buffer (50 mM Tris-HCl, pH 8.0, 300 mM NaCl, and 2 mM phenylmethanesulfonyl fluoride (PMSF)) by a high-pressure homogenizer (800–1000 bar, 15 min). We next purified the over-expressed proteins by using HisTrap HP (5 ml) and HiLoad 16/600 Superdex 200 columns following the manufacturer's instructions (GE Healthcare). The purified proteins were finally in a buffer of 50 mM Hepes-KOH, pH 8.0, 150 mM KCl, 10 mM MgCl$_2$, and 5% glycerol, concentrated, flash frozen in liquid nitrogen, and stored at −80°C. The purity was assessed by SDS-PAGE. Protein concentration was determined by BCA assay (Thermo Fisher).

Human Tau23/K19 was expressed and purified on the basis of a previously described method (*Barghorn et al., 2005*). Briefly, Tau23/K19 was purified by a HighTrap HP SP (5 ml) column (GE Healthcare), followed by a Superdex 75 gel filtration column (GE Healthcare). For $^{15}$N- or $^{15}$N/$^{13}$C-labeled proteins, protein expression was the same as that for unlabeled proteins except that the cells were grown in M9 minimal medium with $^{15}$NH$_4$Cl (1 g l$^{-1}$) in the absence or presence of $^{13}$C$_6$-glucose (2 g l$^{-1}$).

## In vitro tau phosphorylation

Phosphorylation of Tau23/K19 by MARK2 kinase was carried out following a method described previously (*Schwalbe et al., 2013*). Briefly, Tau23/K19 was incubated with cat MARK2-T208E (a hyperactive variant) (*Timm et al., 2003*) at a molar ratio of 10:1 in a buffer of 50 mM Hepes, pH 8.0, 150 mM KCl, 10 mM MgCl$_2$, 5 mM ethylene glycol tetraacetic acid (EGTA), 1 mM PMSF, 1 mM dithiothreitol (DTT), 2 mM ATP (Sigma), and protease inhibitor cocktail (Roche) at 30℃ overnight. Phosphorylated Tau23/K19 was further purified by HPLC (Agilent) to remove kinase, and lyophilized. The sites and degrees of phosphorylation were quantified using 2D $^1$H-$^{15}$N HSQC spectra according to previously published procedures (*Eliezer et al., 2005*; *Schwalbe et al., 2013*; *Timm et al., 2003*).

## Thioflavin T (ThT) fluorescence assay

Amyloid fibril formation of pK19 and pTau23 were monitored using an in situ ThT-binding assay. The ThT kinetics for amyloid fibrils were recorded using a Varioskan Flash Spectral Scanning Multimode Reader (Thermo Fisher Scientific) with sealed 384-microwell plates (Greiner Bio-One). Client proteins were mixed in the absence or presence of NMNATs and variants in indicated molar ratios in a buffer of 50 mM Tris-HCl, 50 mM KCl, 5% glycerol, 0.05% NaN$_3$, pH 8.0, respectively. A final concentration of 50 µM ThT was added to each sample. To promote the formation of amyloid fibrils, 5% (v/v) of fibril seeds (the seeds were prepared by sonicating fibrils for 15 s) were added to pK19 and pTau23, respectively. ThT fluorescence was measured in triplicates with shaking at 600 *rpm* at 37 ℃ with excitation at 440 nm and emission at 485 nm.

## Transmission electron microscopy (TEM)

5 µl of samples were applied to fresh glow-discharged 300-mesh copper carbon grids and stained with 3% v/v uranyl acetate. Specimens were examined by using Tecnai G2 Spirit TEM operated at an accelerating voltage of 120 kV. Images were recorded using a 4K × 4K charge-coupled device camera (BM-Eagle, FEI Tecnai).

## Nuclear magnetic resonance (NMR) spectroscopy

All NMR samples were prepared in an NMR buffer of 25 mM HEPES, 40 mM KCl, 10 mM MgCl$_2$, and 10% (v/v) D$_2$O at pH 7.0. NMR experiments were collected at 298 K on Bruker Avance 600 and 900 MHz spectrometers. Both spectrometers are equipped with a cryogenic TXI probe. Backbone assignments of K19 and pK19 were accomplished according to the previously published assignments (*Eliezer et al., 2005*) and validated by the collected 3D HNCACB and CBCACONH spectra, respectively. These experiments were performed using a ~ 1 mM 15N/13C labeled sample. For HSQC titration experiments, each sample (500 µl) was made of 0.1 mM 15N labeled protein (K19/pK19/Tau/pTau), in the absence or presence of mN3 at a molar ration of 1:2. All NMR spectra were processed using NMRPipe and analyzed using Sparky (*Lee et al., 2015*) and NMRView (*Johnson, 2004*).

## Biolayer interferometry (BLI) assay

The binding affinity between mN3 and client proteins was inspected by BLI experiments with ForteBio Octet RED96 (Pall ForteBio LLC) (*Rich and Myszka, 2007*). All data were collected at 25 ℃ with orbital shaking at 1,000 rpm in 96-well black flat-bottom plates (Greiner Bio-One). A total volume of 200 µl was used for each sample and all reagents were diluted in a buffer of 50 mM HEPES, 150 mM KCl, 10 mM MgCl$_2$ at pH 8.0. Biotinylated mN3 or Hsp90 (20 µg ml$^{-1}$) was loaded onto SA sensors (ForteBio) for 180 s, followed by a 60 s baseline, and then associated with different concentrations of client proteins for 360 s. The association step was followed by a 360 s dissociation step. All data were processed by data analysis software 9.0 (ForteBio). The competition of NMN with pK19 was

monitored with the addition of different concentrations of NMN in both association and dissociation solution. For the association step, 50 µM pK19 was pre-mixed with different concentrations of NMN.

To calculate the EC50 of NMN as a competitor for mN3's interaction with pTau, the BLI data were fit to the equation as described (*Lian et al., 2013*; *Xiao et al., 2019*):

$$B = \frac{EC_{50}^n}{EC_{50}^n + [I]^n}$$

where B represents the response percentage, [I] represents the concentration of NMN used as a competitor, EC50 represents the concentration of NMN that causes a 50% reduction in the BLI response, and n represents the pseudo-Hill coefficient. The EC50 value of the solution competition is 501 µM, n is 0.63, $R^2 = 0.994$.

## mN3 crystallization, data collection and structure determination

Crystals of mN3 were obtained by the hanging drop vapor diffusion method at 18°C. The condition of 0.04 M citric acid, 0.06 M Bis-Tris propane, pH 6.0–7.5, 20% PEG3350 yielded the diffraction quality crystals after 2 days. Before data collection, crystals were soaked in a cryoprotectant solution consisting of the reservoir solution and 10% (v/v) glycerol and then quickly frozen with liquid nitrogen.

Diffraction data of mN3 was collected at the wavelength of 0.9791 Å using an ADSC Quantum 315 r detector at beamline BL17U of Shanghai Synchrotron Radiation Facility (SSRF). Diffraction data for the crystal was collected at 2.00 Å resolutions, as shown in *Supplementary file 1*. The intensity sets of the mN3 crystal was indexed, integrated and scaled with the HKL2000 package (*Otwinowski and Minor, 1997*).

The mN3 structure was solved by molecular replacement method using Phaser (*McCoy et al., 2007*) in the CCP4 crystallographic suite (*Potterton et al., 2004*) with the crystal structure of NMN/NaMN adenylyltransferase (1KQN) as a template. Several cycles of refinement were carried out using Phenix and Coot (*Emsley and Cowtan, 2004*; *Vagin et al., 2004*) progress in the structural refinement was evaluated by the free R-factor.The mN3 structure belong to the P21 space group with cell dimensions a = 53.7 Å, b = 80.8 Å, c = 64.5 Å.

## Size exclusion chromatography and multi-angle laser light scattering (SEC-MALS)

The weight-average molecular weight (Mw) of mN3 was estimated by SEC-MALS that consisted of an SEC column (KD–806 M, Shodex, Tokyo, Japan), a MALS detector (DAWN HELEOS-II,=658 nm, Wyatt Technologies, USA), and a RI detector (Optilab,=658 nm, Wyatt Technologies, USA). 100 ul mN3 (10 mg/ml) in a buffer of 50 mM Hepes-KOH, pH 8.0, 150 mM KCl, 10 mM MgCl$_2$, 5% glycerol and 0.05% NaN$_3$ were loaded to the SEC column with a flow rate of 0.5 ml/min at 25 °C. The resulting data was analyzed using ASTRA VI software (Wyatt Technologies, USA).

## Cross-linking mass spectrometry analysis (XL-MS)

Cross-linking experiments were performed as described previously (*Zhou et al., 2002*). pK19 was incubated with mN3 at 6:1 molar ratio in a buffer containing 50 mM Hepes-KOH, 150 mM KCl at pH 8.0 for 20 min at 4 °C. Cross-linker BS$^3$ (Thermo Fisher Scientific, 21585) was added at a 1:8 mass ratio and incubated at room temperature for 1 hr. The reaction was quenched with 20 mM ammonium bicarbonate at room temperature for 20 min. Cross-linking products were analyzed by SDS-PAGE to assess the cross-linking efficiency. For MS analysis, proteins were precipitated with acetone; the pellet was resuspended in 8 M urea, 100 mM Tris (pH 8.5) and digested with trypsin at 37 °C overnight. The resulting peptides were analyzed by online nanoflow liquid chromatography tandem mass spectrometry (LC-MS/MS). And the mass spectrometry data were analyzed by pLink (*Yang et al., 2012*).

## Differential scanning fluorimetry (DSF)

Thermal melting experiments were carried out using a QuantStudio 6 and 7 Flex Real-Time PCR Systems (Life) as described previously (*Niesen et al., 2007*). The buffer is 50 mM HEPES, 150 mM KCl, 10 mM MgCl$_2$, and 5% glycerol at pH 8.0. SYPRO Orange (Thermo Fisher) was added as a fluorescence probe at a dilution of 1:1000. 10 µL of protein mixed with SYPRO Orange (Thermo Fisher)

solution (1:1000, 50 mM HEPES, 150 mM KCl, 10 mM MgCl$_2$, and 5% glycerol at pH 8.0) to a final concentration of 10 µM were assayed in 384-well plates (Life). Excitation and emission filters for SYPRO-Orange dye were 465 nm and 590 nm, respectively. The temperature was increased by 0.9°C per minute from 25°C to 96°C. The inflection point of the transition curve (Tm) is calculated using protein thermal shift software v1.2 (Thermo Fisher).

## Modeling of the complex structure of peptide RVQ(p)SKIG(p)SLDNI and mN3

The 12-amino acid peptide [349]RVQ(p)SKIG(p)SLDNI[360] ((p)S: phosphoserine) was docked into mN3 following the Rosetta FlexPepDock protocol (*Raveh et al., 2011*) in Rosetta software package (*Leaver-Fay et al., 2011*). Firstly, the 12-mer peptide mimic RVQEKIGELDNI, in which the two phosphoserines were replaced by two glutamates, was docked to the crystal structure of mN3 (PDB: 5Z9R). We performed docking simulations with the restrains of two phosphate binding sites identified in the crystal structure of human cytosolic NMN/NaMN adenylyltransferase (PDB code: 1NUS) (*Zhang et al., 2003*). The extended 12-mer peptide mimic was initially placed near the putative phosphate binding site. 5000 models were generated by using FlexPepDock protocol to simultaneously fold and dock the peptide over the receptor surface. In this fold-and-dock step, we imposed the distance restraints to confine the glutamate residues of peptide mimic within the phosphate binding sites identified from the crystal structure. The top models with favorable Rosetta energies and satisfied constraints were selected, and the phosphoserines were modeled back by replacing two phosphoserine mimic residues glutamates. The newly modeled structure was further refined by energy minimization to get rid of potential clash and maintain the identified phosphate binding site. After refinement, the top models ranked by Rosetta energies and constraints were selected for visual inspection.

## Enzyme activity assay

Enzyme activity of NMNAT was measured in a continuous spectrophotometric coupled assay by monitoring the increase in absorbance of NADH at 340 nm, The reaction process is as follows *Balducci et al. (1995)*:

$$\text{NMN} + \text{APP} \xrightarrow{\text{NMNAT}} \text{NAD}^+ + \text{ppi}$$
$$\text{NAD} + \text{ethnol} \xrightarrow{\text{ADH}} \text{NADH} + \text{H}^+ + \text{acetaldehyde}$$

The reaction solution contains 28 mM HEPES buffer (pH 7.4), 11.2 mM MgCl2, 16 nM semicarbazide-HCl, 0.046 mM ethanol, 1.5 mM ATP, and 0.03 mg/ml yeast alcohol dehydrogenase (Sigma, A7011), and NMNAT or variants. The reaction was initiated by adding NMN to a final concentration of 0.625 mM. All measurements were performed at 37 °C. The activity was calculated using the equation below.

$$\text{Eunit/mg} = \frac{\Delta\text{A}_{340}\text{nm/min} \times \text{V}_{\text{reaction}}}{\text{Co}\beta - \text{NADH} \times \text{V}_{\text{enzyme}} \times [\text{enzyme}]}$$

Where C$_0$β-NADH, the extinction coefficient of β-NADH at 340 nm, is 6.22 (*Zhai et al., 2006*).

## *Drosophila* stocks and genetics

Flies were maintained on cornmeal-molasses-yeast medium at 25 °C, 65% humidity, 12 hr light/dark cycle. The following strains were used in this study: *UAS-Tau$^{WT}$* and *UAS-Tau$^{R406W}$* obtained from Dr. Mel Feany (*Ali et al., 2012*); *UAS-mitoGFP* obtained from Dr. Hugo J. Bellen (Duncan Neurological Research Institute, Baylor College of Medicine); *UAS-Nmnat-PD*, *UAS-Nmnat-PC*, *UAS-Nmnat-PD$^{WR}$* generated from the lab (*Ruan et al., 2015*; *Zhai et al., 2006*), *GMR-GAL4*, *OK371-GAL4*, and *UAS-LifeAct-GFP* obtained from Bloomington Stock Center.

## Immunohistochemical staining of fly brains and salivary glands

Fly brains with attached lamina were dissected as previously described (*Brazill et al., 2018*). Salivary glands were dissected from wandering third-instar larvae (L3). Samples were fixed in freshly made 4% formaldehyde for 15 min, washed in PBS containing 0.4% (v/v) Triton X-100 (PBTX), and

incubated with primary antibodies at 4°C overnight. Samples were then washed with PBTX and incubated with secondary antibodies at room temperature for 2 hr. After that, samples were stained with 4′,6-diamidino-2-phenylindole (DAPI; Thermo Fisher Scientific, Carlsbad, CA, USA) for 10 min and mounted with VECTASHIELD Antifade Mounting Medium (Vector Laboratories Inc, Burlingame, CA, USA). Samples were kept at 4°C until imaging. The following antibodies were used in this study: anti-Brp (1:250, Developmental Studies Hybridoma Bank, East Iowa City, IA, USA), anti-pTau$^{Ser262}$ (1:250, Santa Cruz Biotechnology, CA, USA), anti-pTau$^{Ser202/Thr205}$ (AT8, 1:250, Thermo Fisher Scientific, Carlsbad, CA, USA), and anti-*Drosophila* Nmnat (1:500; *Zhai et al., 2006*).

## Confocal imaging and processing

Fly brains were imaged using an Olympus IX81 confocal microscope coupled with a 60 × oil immersion objective. Images were processed using FluoView 10-ASW software (Olympus) and analyzed using Fiji/Image J (version 1.52). Statistical analyses were performed using Graphpad Prism (version 7.04).

## Western blot analysis

Ten heads of each genotype were homogenized in radioimmunoprecipitation assay (RIPA) buffer (Sigma-Aldrich, St. Louis, MO, USA). Extracted protein samples were mixed with Laemmli sample buffer containing 2% SDS, 10% glycerol, 62.5 mM Tris-HCl (pH 6.8), 0.001% bromophenol blue, and 5% β-mercaptoethanol, and heated at 95°C for 10 min. Proteins were separated by SDS-polyacrylamide gel electrophoresis and transferred to a nitrocellulose membrane. After blocking at room temperature for 1 hr, the membrane was incubated with primary antibody at 4°C overnight, followed by the secondary antibody for 1 hr at room temperature. Imaging was performed on an Odyssey Infrared Imaging system (LI-COR Biosciences) and analyzed using Image Studio (v4.0). Primary antibody dilutions were used as follows: anti-cleaved caspase-3 (Asp175, 1:1,000, Cell Signaling) and anti-β-actin (1:10,000, Sigma-Aldrich).

## Negative geotaxis assay

Groups of 10 age-matched female flies of each genotype were placed in a vial marked with a line drawn horizontally 8 cm above the bottom surface. Flies were given 45 min to fully recover from $CO_2$ anesthesia and were gently tapped onto the bottom of the vial and given 10 s to climb. Flies that passed the 8 cm line were counted. The assay was repeated 10 times, and 10 independent groups (n = 10, a total of 100 flies) of each genotype were tested. To eliminate observer-expectancy bias, the assay was carried out with the examiner masked to the group assignment.

## Single-molecule pull-down (SMPull) assay by TIRF microscopy

Purified pTau23/Tau23 proteins were mixed with a 3-fold Alexa Fluor 647 (Thermo Fisher, A32757) in a reaction buffer (50 mM $NaH_2PO_4$/$Na_2HPO_4$ at pH 7.4, 150 mM KCl, 0.5 mM TCEP) at 37 °C for 1 hr. The labeled proteins were further purified using the Superdex 200 columns (GE Healthcare, USA) in a buffer containing 50 mM $NaH_2PO_4$/$Na_2HPO_4$ at pH 7.4, 150 mM KCl, 0.5 mM TCEP.

All single-molecule assays were performed in the working buffer including 50 mM NaCl, 50 mM Tris, pH 8.0 and 0.1 mM TCEP at room temperature. Single-molecule imaging was conducted in the working buffer containing an oxygen scavenging system consisting of 0.8 mg/ml glucose oxidase, 0.625% glucose, 3 mM Trolox and 0.03 mg/ml catalase to minimize photobleaching. Slides were firstly coated with a mixture of 97% mPEG and 3% biotin-PEG, flow chambers were assembled using strips of double-sided tape and epoxy. Neutravidin and 20 nM biotin-NTA (Biotium) charged with $NiCl_2$ sequentially flew into the flow chamber and each was incubated for 5 min in the working buffer. The immobilization of Hsp90 (5 nM) was mediated by surface-bound $Ni^{2+}$. Next, 4 nM Tau23/pTau23 and various concentrations of mN3 were added and incubated with the immobilized Hsp90 for 10 min before data acquisition. An objective type total internal reflection fluorescence (TIRF) microscopy was used to acquire single-molecule data. Alexa647 labeled Tau23 or pTau23 was excited at 647 nm with a narrow band-pass filter (ET680/40 from Chroma Technology). Single-molecule analysis was performed using software smCamera. Mean spot per image (imaging area 2500 $\mu m^2$) and standard deviation were calculated from 10 different regions.

## Acknowledgements

We thank Dr. Zhijun Liu, Dr. Songzi Jiang and other staff members of the National Center for Protein Science Shanghai for assistance in NMR data collection. This work was supported by the National Natural Science Foundation (NSF) of China (91853113 to D Li and C Liu), the Major State Basic Research Development Program (2016YFA0501902 to C Liu), the Science and Technology Commission of Shanghai Municipality (18JC1420500to C Liu), Shanghai Pujiang Program (18PJ1404300to D Li), the "Eastern Scholar" project supported by Shanghai Municipal Education Commission (to D Li), Shanghai Municipal Science and Technology Major Project (2019SHZDZX02to C Liu), and Innovation Program of Shanghai Municipal Education Commission(2019-01-07-00-02-E00037to D Li). The support also comes from Dr. John T Macdonald Foundation (to C Li), the Lois Pope LIFE fellows Program (to C Li and Y Zhu), NIH grant R56NS095893 and R61AT010408 (to RG Zhai).

## Additional information

### Funding

| Funder | Grant reference number | Author |
|---|---|---|
| National Natural Science Foundation of China | 91853113 | Cong Liu<br>Dan Li |
| Major State Basic Research Development Program | 2016YFA0501902 | Cong Liu |
| Science and Technology Commission of Shanghai Municipality | | Cong Liu |
| Dr. John T. MacDonald Foundation | | Chong Li |
| University of Miami | Lois Pope LIFE fellows Program | Yi Zhu<br>Chong Li |
| National Institutes of Health | R56NS095893 | Rong Grace Zhai |
| National Institutes of Health | R61AT010408 | Rong Grace Zhai |
| Shanghai Pujiang Program | 18PJ1404300 | Dan Li |
| Shanghai Municipal Science and Technology Major Project | 2019SHZDZX02 | Cong Liu |
| Shanghai Municipal Education Commission | Innovation Program 2019-01-07-00-02-E00037 | Dan Li |
| Shanghai Municipal Education Commission | "Eastern Scholar" project | Dan Li |

The funders had no role in study design, data collection and interpretation, or the decision to submit the work for publication.

### Author contributions

Xiaojuan Ma, Data curation, Validation, Methodology; Yi Zhu, Jingfei Xie, Validation, Visualization, Methodology; Jinxia Lu, Jiaqi Liu, Houfang Long, Validation; Chong Li, Jiali Qiang, Shuai Dou, Yi Xiao, Validation, Methodology; Woo Shik Shin, Software, Validation; Chuchu Wang, Chunyu Jia, Juntao Yang, Methodology; Yanshan Fang, Lin Jiang, Yaoyang Zhang, Shengnan Zhang, Supervision; Rong Grace Zhai, Cong Liu, Supervision, Project administration; Dan Li, Supervision, Funding acquisition, Project administration

### Author ORCIDs

Xiaojuan Ma (iD) https://orcid.org/0000-0003-2682-0501
Yi Zhu (iD) https://orcid.org/0000-0002-1778-8880
Chuchu Wang (iD) https://orcid.org/0000-0003-2015-7331
Yanshan Fang (iD) http://orcid.org/0000-0002-4123-0174

Lin Jiang [ID] http://orcid.org/0000-0003-3039-1877
Rong Grace Zhai [ID] https://orcid.org/0000-0002-7599-1430
Cong Liu [ID] https://orcid.org/0000-0003-3425-6672
Dan Li [ID] https://orcid.org/0000-0002-1609-1539

## Decision letter and Author response

Decision letter https://doi.org/10.7554/eLife.51859.sa1
Author response https://doi.org/10.7554/eLife.51859.sa2

## Additional files

### Supplementary files

- Supplementary file 1. Data collection and structure refinement statistics of mN3.
- Supplementary file 2. Cross-linked peptides between pK19 and mN3.
- Transparent reporting form

### Data availability

Diffraction data have been deposited in PDB under the accession code 5Z9R.

The following dataset was generated:

| Author(s) | Year | Dataset title | Dataset URL | Database and Identifier |
| --- | --- | --- | --- | --- |
| Ma X, Dou S, Li D, Liu C | 2019 | NMNAT as a specific chaperone antagonizing pathological condensation of phosphorylated tau | http://www.rcsb.org/structure/5Z9R | RCSB Protein Data Bank, pdb5Z9R |

The following previously published dataset was used:

| Author(s) | Year | Dataset title | Dataset URL | Database and Identifier |
| --- | --- | --- | --- | --- |
| Zhang X, Kurnasov OV, Karthikeyan S, Grishin NV, Osterman AL, Zhang H | 2003 | CRYSTAL STRUCTURE OF HUMAN CYTOSOLIC NMN/NaMN ADENYLYLTRANSFERASE COMPLEXED WITH ATP ANALOG AND NMN | https://www.rcsb.org/structure/1NUS | RCSB Protein Data Bank, 1NUS |

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
