## [Decision Letter]

**Acceptance summary:**

Your work establishes that NMNAT can function as a chaperone for phosphorylated tau both in vitro and in vivo, resulting in reduced aggregation. It establishes a connection between NAD^+^ metabolism and tau homeostasis. This is a novel mechanism that may well have implications for the understanding of human tauopathies.

**Decision letter after peer review:**

Thank you for submitting your article "Mechanistic insights into NAD synthase NMNAT chaperoning phosphorylated Tau from pathological aggregation" for consideration by *eLife*. Your article has been reviewed by three peer reviewers, and the evaluation has been overseen by a Reviewing Editor and David Ron as the Senior Editor. The following individuals involved in review of your submission have agreed to reveal their identity: Hui-Chen Lu (Reviewer #1); Jan Bieschke (Reviewer #2); Corinne Lasmezas (Reviewer #3).

The reviewers have discussed the reviews with one another and the Reviewing Editor has drafted this decision to help you prepare a revised submission.

The reviewers are positive about the manuscript, but they raised a number of concerns that should be addressed.

1) In particular, please examine in detail the interaction between NMNAT and tau and the modulating effect of the KKRK mutation in the fly model. It is important that you establish that the biophysical principles that govern the interaction in vitro, also do so in the tissues of animals, in which the functional impact of NMNAT is observed (refers to reviewer 1, point 2 and reviewer 2, point 1).

2) Please address the plausibility of NMNAT in exploiting the same biophysical principles to engage phosphorylated Tau and HTT (see reviewer 2, point 1).

3) Finally, as the Editor overseeing this review process, I'd like to chime in with a request that you clarify the role of NMN as competitor for NMNAT's interaction with pTau (as revealed in Figure 3C) by performing an formal analysis of NMN's Ki for this interaction and discussing any observation thus made in the context of what is known about the concentration of this metabolite in cells.

The full reviews are included below for your reference and we would be grateful if you could address these when revising your manuscript.

Reviewer #1:

In this study, Ma et al. reveal novel insights on how the enzymatic pocket critical for NAD synthesis of NMNATs is also important to reduce aggregation of phosphorylated Tau (pTau). A series of elegant structural analyses including NMR, crystallography, single-molecule and computational approaches were employed. The authors investigated NMNAT's chaperone activity against amyloid aggregation by combining biophysical and computational approaches and provide a structural basis for NMNAT's enzymatic and chaperone activity. Specifically, NMR data show the specific difference in chemical shifts of phosphorylated serine residues. A good sign for the specificity of this sidechain interaction with mN3 is evident in that the rest of the residues of the pK19 structure do not shift. In addition, they also showed how NMNAT3 serves as co-chaperone of Hsp90 for pTau clearance. The same can be said for the supplementary results showing NMR data for Tau23. A positive of this study is the great statistical analysis increasing its scientific rigors.

1) The authors failed to address whether mN3 acts as a functional dimer or monomer. Since their SEC-MALS data indicate a dimer in solution, the structure and function should be evaluated in the context of a dimer and not necessarily just in the context of a monomer. Additional SEC-MALS data could provide clues whether mN3 forms concentration-dependent dimers. It is critical to show SEC-MALS for a range of concentrations of mN3. Creating a monomer only mN3 mutant and repeating the binding and chaperone assays would be valuable to test if enzymatic versus chaperone function is mN3 dimer dependent or whether the common pTau and ATP/NMN binding site of the mN3 monomer is sufficient. This could be tested by disruption the dimer interface, create a monomer-only mutant and check whether it still has both, enzymatic and chaperone functions and whether KKRK mutation disrupt the binding with pTau. Alternatively, authors should revise and clarify their findings are based on dimers.

2) For in vivo testing with the humanized *Drosophila* tauopathy model, the authors should test if KKRK mutations alter NMNAT's pTau clearance capability. The current data for the clearance of Tau aren't particularly novel based on previous publications (e.g. Ali et al., 2012 and Ljungberg et al., 2012). The in vivo demonstration with KKRK mutant will improve the novelty of their major conclusion.

Reviewer #2:

The authors characterize a new anti-amyloid activity of the enzyme nicotinamide mononucleotide adenylyltransferase (NMNAT). They find that the enzyme potently inhibits fibril formation by phosphor-tau protein, which is deposited in neurofibrillary tangles in Alzheimer's disease. Furthermore NMNAT upregulation ameliorates pTau toxicity in a *Drosophila* model for tau deposition. The findings in the fly model mirror the anti-amyloid effect of NMNAT against Huntingtin toxicity and aggregation in *Drosophila*, which the same author team just published (Zhu et al., 2019). Here the authors first analyze the mechanism of inhibition in vitro via Thioflavin binding and EM and find that the enzyme inhibits tau fibril formation and causes the formation of smaller aggregates instead. Chemical shift analysis shows that both proteins bind, likely in their monomeric forms, and that pSer residues of tau are directly affected by NMNAT binding. Mass spectrometry combined with chemical crosslinking identified residues in the substrate binding pocket of NMNAT to interact with tau. Mutation of positively charged lysines to glutamates abolished binding. This suggests a specific substrate binding to phosphor-tau via charge interactions.

The finding that NMNAT, in complex with Hsp90, possesses direct chaperone-like anti-amyloid activity is exciting and may provide new therapeutic avenues to Alzheimer and other protein misfolding diseases. The in vitro data in the manuscript are convincing and comprehensive.

1) My main point of criticism centers on the question whether the conclusions drawn in vitro actually apply to the animal (and human) system. This is notoriously difficult to answer, but a reasonably straightforward test would be the co-immunostaining of pTau and NMNAT (PD) in the fly images. My reason for concern is that the proposed mechanism for pTau inhibition seems slightly at odds with the inhibition of Huntingtin aggregation by NMNAT observed by the same authors. Htt is not known to be associated with hyperphosphorylation, poly-glutamine sequences are uncharged and N-terminal phosphorylation actually inhibits Htt aggregation (Deguire, JBC 2018). I am aware that the PNAS paper had not yet been published at the time of manuscript submission, but an updated version of this manuscript needs to address this apparent mechanistic discrepancy between both systems and provide evidence that both proteins interact in vivo.

2) Does NMNAT only inhibit amyloid formation by competition for the substrate protein, i.e. pTau, or is there also a chaperone-like effect on tau that is already misfolded/aggregated? The authors show ample evidence for the first mechanism, however, I hesitate to call a protein a chaperone that requires quantitative complex formation to inhibit misfolding. It would be interesting to see if the enzyme also inhibited seeded fibril growth, or even disassembled amyloid fibrils of pTau. Related, the authors find increased interaction between pTau and the chaperone HSP90 in the presence of NMNAT. Is this interaction between the monomeric protein or an aggregate?

3) The dissociation constants for the enzyme to pTau K19 are ~30 μM. How does this compare to the concentrations of both proteins in vivo? The authors observed that the presence of NMNAT substrates strongly inhibited anti-amyloid activity. Here, the authors should discuss the relative concentrations in vivo and their implications for anti-amyloid activity.

4) Crosslinking experiments show liking to two tau regions around a.a. 350 and a.a. 250. Only the 350 region is shown in the binding model and is discussed in the context of NMR data. Does binding to the a.a. 250 region not contribute to the anti-amyloid effect?

---

## [Author Response]

The reviewers are positive about the manuscript, but they raised a number of concerns that should be addressed.1) In particular, please examine in detail the interaction between NMNAT and tau and the modulating effect of the KKRK mutation in the fly model. It is important that you establish that the biophysical principles that govern the interaction in vitro, also do so in the tissues of animals, in which the functional impact of NMNAT is observed (refers to reviewer 1, point 2 and reviewer 2, point 1).

To characterize the interaction between Nmnat and pTau in vivo, we followed the reviewer’s suggestion and performed a coimmunostaining of pTau and Nmnat. We took advantage of the *Drosophila* salivary gland cells due to their big size that allows optimal resolution for colocalization analysis. We used *OK371-GAL4* to express human Tau^R406W^ and Nmnat(PD), dissected the salivary glands at the third instar larval stage, and co-stained pTau and Nmnat. We observed a clear colocalization of Nmnat with pTau in the cytoplasm (new Figure 4—figure supplement 4), indicating an interaction between Nmnat and pTau in vivo. In addition, we also generated a Nmnat^WR^ mutant with abolished enzymatic activity but only slightly decreased chaperone activity (new Figure 4—figure supplement 3). We found that Nmnat^WR^ also colocalized with pTau in vivo(new Figure 4—figure supplement 4).

It would be interesting to test the in vivo function of the KKRK mutant. However, the analysis on KKRK mutant may have the following difficulties. First, although the KKRK residues are generally conserved across difference species, the counterpart residues in *Drosophila* are not exactly the same. The corresponding residue for mN3 K206 in *Drosophila* is R. Second, since KKRK mutations abolished the both enzymatic and chaperone activities of mN3 (new Figure 4—figure supplement 3), we would expect a complete lack of neuroprotection in vivo. Therefore, it is difficult to pinpoint whether the loss of neuroprotection is due to the lack of enzymatic activity or chaperone activity.

To address this question and overcome these concerns, we took advantage of a previously established *Drosophila* line with *UAS-Nmnat^WR^* (W98G/R224A double mutant) insertion that can be used to express enzyme-inactive Nmnat (Zhai, et al., 2006). Our previous study has shown that although Nmnat^WR^ still confers protection in a *Drosophila* model of tauopathy, its protective capacity is significantly reduced compared to wild type Nmnat protein, especially at an advanced disease stage (Ali et al., 2012). Although noted, this difference has been puzzling until the structural study carried out in the present report. As the W98G/R224A double mutation include the R224 residue in the ISSTXXR motif, the same R residue as in the KKRK mutant, the WR mutant is expected to have reduced binding to pTau. Therefore, the Nmnat^WR^ line will be a useful tool to allow careful dissection of the contribution of individual residues in binding hyperphosphorylated Tau and in neuroprotection. It is also important to note that in recent years we have dedicated significant efforts to develop new quantitative methods to improve the resolution of our biochemical and imaging analyses of neuroprotection, which makes a careful analysis on the contribution of individual residues possible.

In the revised manuscript, we first carried out in vitro analyses on Nmnat^WR^ mutant protein. We found that compared to wild type Nmnat, Nmnat^WR^ has abolished enzymatic activity and slightly decreased chaperone activity (new Figure 4—figure supplement 3). We then used *GMR-GAL4* to express *Drosophila* Nmnat^WR^ with Tau^WT^ or Tau^R406W^ in the photoreceptors and characterized the in vivo cellular consequences including mitochondrial phenotype (revised Figure 4, new Figure 4—figure supplement 1) and F-actin phenotype (revised Figure 5). We found that compared to wild type Nmnat, Nmnat^WR^ shows slightly increased mitoGFP clustering at the lamina cortex, reduced mitoGFP and increased F-actin accumulation at R7 and R8 terminals (revised Figure 4, new Figure 4—figure supplement 1; revised Figure 5), indicating that Nmnat^WR^ has modestly reduced protective capacity against tauopathy in vivo, consistent with the conclusion that the R residue in the ISSTXXR adenylyltransferase is involved in chaperoning pTau. Notably, we did not find a significant difference in pTau level when overexpressing wild type Nmnat and Nmnat^WR^ (new Figure 4—figure supplement 1C), indicating that the slightly decreased protective capacity of Nmnat^WR^ is unlikely due to an alteration of pTau clearance capability.

Taken together, our in vitro and in vivo experiments show that: (1) both Nmnat and Nmnat^WR^ interact and chaperone pTau from pathological aggregation; (2) the protection of Nmnat in tauopathy not only comes from the clearance of pTau but also a reduction of pTau cytotoxicity and improvement of neuronal integrity; and (3) the Nmnat^WR^ mutant with reduced binding to pTau confers proportionally reduced protection in vivo, suggesting the structural overlap between the chaperone activity and enzymatic activity.

To further improve the novelty and significance of our study, we performed a comprehensive analysis of the isoform-specific protective capacity of Nmnat in vivo (Ali et al., 2012). We have shown that the *Drosophila* Nmnat gene is alternatively spliced and produced two protein isoforms: a nuclear isoform PC and a cytoplasmic isoform PD (Ruan, et al., 2015). Our in vitrodata have shown that both Nmnat isoforms PC and PD exhibited a potent chaperone activity against amyloid aggregation of pTau23 and pK19 (Figure 1—figure supplement 3 and Figure 1—figure supplement 4). However, these two isoforms exhibit distinct protective capacities in vivo. While PD shows potent protection against tauopathy as evidenced by reduced pTau level, reduced brain apoptosis, and improved locomotor activity, PC has minimal protective capacity (revised Figure 5—figure supplement 1 and new Figure 5—figure supplement 2). Therefore, the protective capacity of Nmnat in vivo is not only dependent on its chaperone function but also its subcellular localization, as the cytoplasmically localized PD isoform is likely more accessible to the client pTau protein.

2) Please address the plausibility of NMNAT in exploiting the same biophysical principles to engage phosphorylated Tau and HTT (see reviewer 2, point 1).

To address this concern, we would like to note that without phosphorylation, mN3 still shows interactions with the KXGS motifs of Tau (Figure 1E, Figure 1—figure supplement 6B), although the interaction of mN3 to Tau is weaker than that to pTau. Based on the mN3 structure, there is a hydrophobic area on the periphery of the phosphate-binding site (Figure 2B), which implies that hydrophobic interactions may also contribute to the binding of NMNAT to client proteins. Recent study showed that NMNAT can also interact with Htt (Zhu et al., 2019). Htt is uncharged and not known to be associated with hyperphosphorylation. Thus, NMNAT may have a wide-spectrum anti-amyloid activity with versatile mechanisms. Our work suggests that electrostatic interaction, especially a specific binding between the MARK2 phosphorylation sites of pTau with the enzymatic pocket of NMNAT, can remarkably enhance the binding of NMNAT to Tau.

In the revised manuscript, we added a new paragraph in the Discussion section “NMNAT is distinct from canonical molecular chaperones” to clarify this point.

3) Finally, as the Editor overseeing this review process, I'd like to chime in with a request that you clarify the role of NMN as competitor for NMNAT's interaction with pTau (as revealed in Figure 3C) by performing an formal analysis of NMN's Ki for this interaction and discussing any observation thus made in the context of what is known about the concentration of this metabolite in cells.

We determined the EC50 of NMN as a competitor for mN3’s interaction with pTau. EC50 represents the concentration of NMN that causes a 50% reduction in the BLI response. We fit the data of Figure 3C to the equation as described by Lian et al., 2013 and Xiao et al., 2019, and calculated the EC50 value of the solution competition is 501 μM. We added the EC50 value in the revised Results and the detailed methods of EC50 calculation in the revised Materials and methods, section “Biolayer interferometry (BLI) assay”.

NMN is a key intermediate of NAD^+^, and the basal level of NMN is low. It has been reported that the basal concentration of NMN in mouse DRG neurons is ~0.01 μM (Sasaki et al., e*Life*, 2016). However, NAD^+^ as the final product has a concentration of μM in neurons (Sasaki, et al., e*Life*, 2016). The levels of NMN and NAD^+^ depend on the activity of nicotinamide phosphoribosyltransferase (NAMPT) that converts nicotinamide and PRPP to NMN and is a rate-limiting enzyme in NAD^+^ synthesis (Stein, et al., 2014). It has been shown that NAMPT and NAD^+^ levels decline during aging (Stein, et al., 2014; Gomes, et al., 2013). Therefore, we may assume that in young and healthy neurons, where the pTau level is low and the NAD^+^ synthesis is normal, NMNAT would mainly function to generate NAD^+^. In contrast, in old and degenerating neurons, where pTau increases and NAD^+^ synthesis declines, NMNAT would switch its function to inhibit pTau aggregation and assist Hsp90 for pTau clearance. In the revised manuscript, we added this discussion as a new paragraph in the revised Discussion, section “NMNAT links NAD^+^ metabolism and Tau homeostasis”.Reviewer #1:[…] 1) The authors failed to address whether mN3 acts as a functional dimer or monomer. Since their SEC-MALS data indicate a dimer in solution, the structure and function should be evaluated in the context of a dimer and not necessarily just in the context of a monomer. Additional SEC-MALS data could provide clues whether mN3 forms concentration-dependent dimers. It is critical to show SEC-MALS for a range of concentrations of mN3. Creating a monomer only mN3 mutant and repeating the binding and chaperone assays would be valuable to test if enzymatic versus chaperone function is mN3 dimer dependent or whether the common pTau and ATP/NMN binding site of the mN3 monomer is sufficient. This could be tested by disruption the dimer interface, create a monomer-only mutant and check whether it still has both, enzymatic and chaperone functions and whether KKRK mutation disrupt the binding with pTau. Alternatively, authors should revise and clarify their findings are based on dimers.

We thank the reviewer for this great point and her suggestions. The structures of several NMNAT homologs have been determined and their oligomeric states in solution have been carefully characterized. As shown in Figure 2—figure supplement 1B, similar to mN3, *Bacillus subtilis* NMNAT exists as a dimer in solution and as reported, this dimer is not affected by the concentration decrease (Olland, et al., 2002). Similarly, as we decreased the concentration of mN3, the dimer remains intact by size exclusion chromatography (Figure 2—figure supplement 1D). Unlike mN3 and BsN, hN1 and hN3 exist as tetramer and hexamer in solution, respectively (Figure 2—figure supplement 1B). However, an equilibrium between dimer and hexamer has been observed (Zhou, et al., JBC, 2002), indicating that dimer is a functional unit. Moreover, the structures of these NMNAT proteins show that the functional dimer contains a highly conserved interface (Figure 2—figure supplement 1B). This indicates that this interface (or dimerization) is important for the NMNAT activity. However, the active site is away from the interface, thus the role of dimerization is puzzling. It has been suggested that the dimer interface may play a role in ATP binding since residues proximal to the interface form part of the ATP-binding pocket (Olland, et al., 2002).

To investigate the role of dimerization in mN3 activities, taken the reviewer’s suggestion, we mutated E198P and L217R in the dimer interface (referred to as EL mutant) to disrupt the interface. The result of SEC-MALS showed that the EL mutations partially dissociated the dimerization of mN3 (Figure 2—figure supplement 1E). The dissociation of dimer resulted in a marked decrease of protein stability and enzymatic activity (Figure 2—figure supplement 1F and G). Intriguingly, the anti-amyloid activity appeared not affected (Figure 2—figure supplement 1H). These results indicate that dimerization is important to stabilize the overall structure and the enzymatic pocket. In contrast, anti-amyloid activity requires a less defined pocket than that for the enzymatic activity, and thus less relies on the dimerization.

We agree with the reviewer on the necessity of clarifying the role of mN3 dimerization in its dual activities. Thus, we added Figure 2—figure supplement 1B and Figure 2—figure supplement 1D-H in the revised manuscript, and described these results in the Results, section “mN3 utilizes its enzymatic substrate-binding site to bind pTau” as following:

“The crystal contains two mN3 molecules forming a homo-dimer with a buried surface area of 1,075.8 Å2 in the asymmetric unit (Figure 2—figure supplement 1B). […] This difference indicates different mechanisms of the dual activities of mN3.”

2) For in vivo testing with the humanized *Drosophila* tauopathy model, the authors should test if KKRK mutations alter NMNAT's pTau clearance capability. The current data for the clearance of Tau aren't particularly novel based on previous publications (e.g. Ali et al., 2012 and Ljungberg et al., 2012). The in vivo demonstration with KKRK mutant will improve the novelty of their major conclusion.

We thank the reviewer for this important point. We agree with the reviewer that it would be interesting to test the in vivo function of the KKRK mutant. However, the analysis on KKRK mutant may have the following difficulties. First, although the KKRK residues are generally conserved across difference species, the counterpart residues in *Drosophila* are not exactly the same. The corresponding residue for mN3 K206 in *Drosophila* is R. Second, since KKRK mutations abolished the both enzymatic and chaperone activities of mN3 (see Figure 4—figure supplement 3), we would expect a complete lack of neuroprotection in vivo. Therefore, it is difficult to pinpoint whether the loss of neuroprotection is due to the lack of enzymatic activity or chaperone activity.

To address this question and overcome these concerns, we took advantage of a previously established *Drosophila* line with *UAS-Nmnat^WR^* (W98G/R224A double mutant) insertion that can be used to express enzyme-inactive Nmnat (Zhai et al., 2006). Our previous study has shown that although Nmnat^WR^ still confers protection in a *Drosophila* model of tauopathy, its protective capacity is significantly reduced compared to wild type Nmnat protein, especially at an advanced disease stage (Ali, et al., 2012). Although noted, this difference has been puzzling until the structural study carried out in the present report. As the W98G/R224A double mutation include the R224 residue in the ISSTXXR motif, the same R residue as in the KKRK mutant, the WR mutant is expected to have reduced binding to pTau. Therefore, the Nmnat^WR^ line will be a useful tool to allow careful dissection of the contribution of individual residues in binding hyperphosphorylated Tau and in neuroprotection. It is also important to note that in recent years we have dedicated significant efforts to develop new quantitative methods to improve the resolution of our biochemical and imaging analyses of neuroprotection, which makes a careful analysis on the contribution of individual residues possible.

In the revised manuscript, we first carried out in vitro analyses on Nmnat^WR^ mutant protein. We found that compared to wild type Nmnat, Nmnat^WR^ has abolished enzymatic activity and slightly decreased chaperone activity (new Figure 4—figure supplement 3). We then used *GMR-GAL4* to express *Drosophila* Nmnat^WR^ with Tau^WT^ or Tau^R406W^ in the photoreceptors and characterized the in vivo cellular consequences including mitochondrial phenotype (revised Figure 4, new Figure 4—figure supplement 1) and F-actin phenotype (revised Figure 5). We found that compared to wild type Nmnat, Nmnat^WR^ shows slightly increased mitoGFP clustering at the lamina cortex, reduced mitoGFP and increased F-actin accumulation at R7 and R8 terminals (we have included a new model in revised Figure 5C, indicating that Nmnat^WR^ has modestly reduced protective capacity against tauopathy in vivo, consistent with the conclusion that the R residue in the ISSTXXR adenylyltransferase is involved in chaperoning pTau. Notably, we did not find a significant difference in pTau level when overexpressing wild type Nmnat and Nmnat^WR^ (new Figure 4—figure supplement 1C), indicating that the slightly decreased protective capacity of Nmnat^WR^ is unlikely due to an alteration of pTau clearance capability. Moreover, to examine the interaction between Nmnat and pTau in vivo, we co-expressed Tau^R406W^ and Nmnat/Nmnat^WR^ in third-instar larval salivary gland cells. We found that both wild type Nmnat and Nmnat^WR^ colocalize with pTau (new Figure 4—figure supplement 4), indicating a direct interaction between Nmnat/Nmnat^WR^ and pTau.

Taken together, our in vitro and in vivo experiments show that: (1) both Nmnat and Nmnat^WR^ interact and chaperone pTau from pathological aggregation; (2) the protection of Nmnat in tauopathy not only comes from the clearance of pTau but also a reduction of pTau cytotoxicity and improvement of neuronal integrity; and (3) the Nmnat^WR^ mutant with reduced binding to pTau confers proportionally reduced protection in vivo, suggesting the structural overlap between the chaperone activity and enzymatic activity.

To further improve the novelty and significance of our study, we performed a comprehensive analysis of the isoform-specific protective capacity of Nmnat in vivo, which was not characterized in our previous study (Ali, et al., 2012). We have shown that the *Drosophila* Nmnat gene is alternatively spliced and produced two protein isoforms: a nuclear isoform PC and a cytoplasmic isoform PD (Ruan et al., 2015). Our in vitrodata have shown that both Nmnat isoforms PC and PD exhibited a potent chaperone activity against amyloid aggregation of pTau23 and pK19 (Figure 1—figure supplement 3 and Figure 1—figure supplement 4). However, these two isoforms exhibit distinct protective capacities in vivo. While PD shows potent protection against tauopathy as evidenced by reduced pTau level, reduced brain apoptosis, and improved locomotor activity, PC has minimal protective capacity (revised Figure 5—figure supplement 1 and new Figure 5—figure supplement 2). Therefore, the protective capacity of Nmnat in vivo is not only dependent on its chaperone function but also its subcellular localization, as the cytoplasmically localized PD isoform is likely more accessible to the client pTau protein.

Based on these new data, we re-wrote the Results, section “NMNAT protects pTau-induced synaptopathy in *Drosophila*”.

Reviewer #2:[…] 1) My main point of criticism centers on the question whether the conclusions drawn in vitro actually apply to the animal (and human) system. This is notoriously difficult to answer, but a reasonably straightforward test would be the co-immunostaining of pTau and NMNAT (PD) in the fly images. My reason for concern is that the proposed mechanism for pTau inhibition seems slightly at odds with the inhibition of Huntingtin aggregation by NMNAT observed by the same authors. Htt is not known to be associated with hyperphosphorylation, poly-glutamine sequences are uncharged and N-terminal phosphorylation actually inhibits Htt aggregation (Deguire, JBC 2018). I am aware that the PNAS paper had not yet been published at the time of manuscript submission, but an updated version of this manuscript needs to address this apparent mechanistic discrepancy between both systems and provide evidence that both proteins interact in vivo.

We thank the reviewer for raising this excellent point. To characterize the interaction between Nmnat and pTau in vivo, we followed the reviewer’s suggestion and performed a coimmunostaining of pTau and Nmnat. We took advantage of the *Drosophila* salivary gland cells due to their big size that allows optimal resolution for colocalization analysis. We used *OK371-GAL4* to express human Tau^R406W^ and Nmnat(PD), dissected the salivary glands at the third instar larval stage, and co-stained pTau and Nmnat. We observed a clear colocalization of Nmnat with pTau in the cytoplasm (new Figure 4—figure supplement 4), indicating an interaction between Nmnat and pTau in vivo. In response to reviewer #1’s comment, we also generated a Nmnat^WR^ mutant with abolished enzymatic activity but only slightly decreased chaperone activity (new Figure 4—figure supplement 3). We found that Nmnat^WR^ also colocalized with pTau in vivo(Figure 4—figure supplement 4), and overexpression of Nmnat^WR^ still confers neuroprotection (revised Figure 4, new Figure 4—figure supplement 1, and revised figure 5).

As for the concern of the proposed mechanism for pTau inhibition, we’d like to note that without phosphorylation, mN3 still shows weak interactions with the KXGS motifs of Tau (Figure 1E, Figure 1—figure supplement 6B). Based on the mN3 structure, there is a hydrophobic area on the periphery of the phosphate-binding site (Figure 2B), which implies that hydrophobic interactions may also contribute to the binding of NMNAT to client proteins. As in the mentioned study (Zhu et al., 2019), NMNAT can also interact with Htt. Thus, NMNAT may have a wide-spectrum anti-amyloid activity, while MARK phosphorylation can specifically enhance the binding of NMNAT to Tau. In the revised manuscript, we added a paragraph in the Discussion as following to clarify this point:

“It is important to point out that although phosphorylation remarkably increases the binding affinity of mN3 to pTau, without phosphorylation, mN3 still shows weak interactions with the KXGS motifs of Tau (Figure 1E, Figure 1—figure supplement 6B). […] Our study indicates both electrostatic and hydrophobic interactions are involved in the NMNAT-client binding, while NMNAT specifically binds to pTau through its enzymatic pocket.”

2) Does NMNAT only inhibit amyloid formation by competition for the substrate protein, i.e. pTau, or is there also a chaperone-like effect on tau that is already misfolded/aggregated? The authors show ample evidence for the first mechanism, however, I hesitate to call a protein a chaperone that requires quantitative complex formation to inhibit misfolding. It would be interesting to see if the enzyme also inhibited seeded fibril growth, or even disassembled amyloid fibrils of pTau. Related, the authors find increased interaction between pTau and the chaperone HSP90 in the presence of NMNAT. Is this interaction between the monomeric protein or an aggregate?

We thank the reviewer’s thoughtful suggestion. Following his suggestion, we performed ThT assays to see if mN3 is able to inhibit seeded fibril growth, or disassemble amyloid fibrils of pTau. The result showed that mN3 efficiently inhibited 8 pK19 fibril formation in the presence of preform fibril seeds (Figure 1—figure supplement 4B). However, mN3 exhibited no disaggregase activity to preformed pK19 fibrils (Figure 1—figure supplement 4C), which was confirmed by TEM microscopy (Figure 1—figure supplement 4C). We agree with the reviewer that NMNAT is different from canonical chaperones in terms of its interacting with pTau. It might be more appropriate to call NMNAT a chaperone-like protein rather than a chaperone. We have revised the manuscript accordingly. As for the NMNAT-mediated interaction between Hsp90 and pTau/Tau by the SMpull assay, we applied monomeric pTau/Tau in the experiment. The reason for using monomer is the recognition of the complex formed by Hsp90 and Tau monomer (Elif Karagoz, et al., Cell, 2014). We clarified the use of pTau/Tau monomer in this experiment in the figure legend of Figure 6 in the revised manuscript.

3) The dissociation constants for the enzyme to pTau K19 are ~30 μM. How does this compare to the concentrations of both proteins in vivo? The authors observed that the presence of NMNAT substrates strongly inhibited anti-amyloid activity. Here, the authors should discuss the relative concentrations in vivo and their implications for anti-amyloid activity.

We thank the reviewer for his suggestion. The dissociation constants for mN3 to pK19 is ~ 31 µM and to pTau23 is ~ 9.9 µM. Tau is a microtubule-binding protein and expressed at the μM level in neurons (Wegmann, the EMBO J., 2018). Importantly, it has been shown that the level of total tau in AD is about eight folds higher than that in normal cases, and this increase is in the form of the abnormally phosphorylated Tau (Khatoon et al., J. Neurochem., 1992). In contrast to the high level of Tau, the expression level of NMNAT in vivo is low. It has been reported that NMNAT2 is a low abundant (nanomolar) protein in neurons (Mayer et al., JBC, 2010). However, NMNAT is not dispersed in neurons. NMNAT2, the cytosolic isoform, is adhesive to Golgi, vesicles and synaptic compartments through post-translational modification and protein-protein interaction (Mayer et al., JBC, 2010). Thus, NMNAT may reach a high local concentration in vivo.

NMN is a key intermediate of NAD^+^, and thus the basal level of NMN is low. It has been reported that the basal concentration of NMN in mouse DRG neurons is ~0.01 μM (Sasaki et al., e*Life*, 2016). However, NAD^+^ as the final product has a concentration of μM in neurons (Sasak, et al., e*Life*, 2016). The levels of NMN and NAD^+^ depend on the activity of nicotinamide phosphoribosyltransferase (NAMPT) that converts nicotinamide and PRPP to NMN and is a rate-limiting enzyme in NAD^+^ synthesis. It has been shown that NAMPT and NAD^+^ levels decline during aging (Steinet al., 2014; Gomes et al., 2013). Therefore, we may assume that in young and healthy neurons, where the pTau level is low and the NAD synthesis is normal, NMNAT would mainly function to generate NAD^+^. In contrast, in old and degenerating neurons, where pTau increases, and NAMPT expression and NAD^+^ production decline, NMNAT would switch its function to chaperone pTau aggregation and clearance. We revised the Discussion and added a paragraph as following:

“NMN is a key intermediate of NAD^+^. The levels of NMN and NAD^+^ depend on the activity of nicotinamide phosphoribosyltransferase (NAMPT) that converts nicotinamide and PRPP to NMN and is a rate-limiting enzyme in NAD^+^ synthesis (Stein et al., 2014). […] In contrast, in old and degenerating neurons, where pTau increases and NAD^+^ synthesis declines, NMNAT would switch its function to inhibit pTau aggregation and assist Hsp90 for pTau clearance (Figure 7).”

4) Crosslinking experiments show liking to two tau regions around a.a. 350 and a.a. 250. Only the 350 region is shown in the binding model and is discussed in the context of NMR data. Does binding to the a.a. 250 region not contribute to the anti-amyloid effect?

Binding to the ~a.a. 250 region of pK19 also contributes to the anti-amyloid effect of mN3. Consistent with the crosslinking data, our NMR titration experiment showed that the intensity attenuation of both ~a.a. 350 and ~a.a. 250 regions of pK19 upon mN3 titration (Figures 1E and 2C). In addition, NMR result also showed interaction of the ~a.a. 320 region of pK19 with mN3, which was not identified by cross-linking (Figures 1E, 2C and Supplementary file 2). The reason for mainly building a model for the complex of mN3 and ~a.a. 350 region of pTau is that the binding of mN3 to ~a.a. 350 region is markedly stronger than to the other two regions (Figures 1E and 2C). While, as mentioned by the reviewer, we should have described the binding of mN3 to the other two regions in the context of the NMR data. Thus, we added this description in the revised Results, section “Mechanism of the interaction between mN3 and pTau”, as following:

“Residues adjacent to pSer, including regions around a.a. 250, a.a. 320 and a.a. 350, also exhibited prominent signal attenuations (Figure 1E)[…] only slight overall signal broadening was observed in the three regions […]”

Also in Results, section “mN3 utilizes its enzymatic substrate-binding site to bind pTau”, following:

“The HSQC spectrum showed that the KKRK mutations significantly diminished the affinity of mN3 to the three regions around a.a. 250, a.a. 320 and a.a. 350 that contain pSer residues (Figure 2C and Figure 2—figure supplement 3). Especially, the region around a.a. 350 (residues 349-360) of R4, which contains two pSer residues […]”